# Deep Neural Tangent Kernel and Laplace Kernel Have the Same RKHS

**Lin Chen**
Simons Institute for the Theory of Computing
University of California, Berkeley
lin.chen@berkeley.edu

**Sheng Xu**
Department of Statistics and Data Science
Yale University
sheng.xu@yale.edu

## Abstract

We prove that the reproducing kernel Hilbert spaces (RKHS) of a deep neural tangent kernel and the Laplace kernel include the same set of functions, when both kernels are restricted to the sphere $\mathbb{S}^{d-1}$. Additionally, we prove that the exponential power kernel with a smaller power (making the kernel less smooth) leads to a larger RKHS, when it is restricted to the sphere $\mathbb{S}^{d-1}$ and when it is defined on the entire $\mathbb{R}^d$.

## 1 Introduction

In the past few years, one of the most seminal discoveries in the theory of neural networks is the neural tangent kernel (NTK) (Jacot et al., 2018). The gradient flow on a normally initialized, fully connected neural network with a linear output layer in the infinite-width limit turns out to be equivalent to kernel regression with respect to the NTK (This statement does not necessarily hold for a non-linear output layer, because the NTK is non-constant (Liu et al., 2020)). Through the NTK, theoretical tools from kernel methods were introduced to the study of deep overparametrized neural networks. Theoretical results were thereby established regarding the convergence (Allen-Zhu et al., 2019; Du et al., 2019b;a; Zou et al., 2020), generalization (Cao & Gu, 2019; Arora et al., 2019b), and loss landscape (Kuditipudi et al., 2019) of overparametrized neural networks in the NTK regime.

While NTK has proved to be a powerful theoretical tool, a recent work (Geifman et al., 2020) posed an important question whether the NTK is significantly different from our repertoire of standard kernels. Prior work provided empirical evidence that supports a negative answer. For example, Belkin et al. (2018) showed experimentally that the Laplace kernel and neural networks had similar performance in fitting random labels. In the task of speech enhancement, exponential power kernels $K_{\exp}^{\gamma,\sigma}(x,y) = e^{-\|x-y\|^\gamma/\sigma}$, which include the Laplace kernel as a special case, outperform deep neural networks with even shorter training time (Hui et al., 2019). The experiments in (Geifman et al., 2020) also exhibited similar performance of the Laplace kernel and the NTK.

The expressive power of a positive definite kernel can be characterized by its associated reproducing kernel Hilbert space (RKHS) (Saitoh & Sawano, 2016). The work (Geifman et al., 2020) considered the RKHS of the kernels restricted to the sphere $\mathbb{S}^{d-1} \triangleq \{x \in \mathbb{R}^d \mid \|x\|_2 = 1\}$ and presented a partial answer to the question by showing the following subset inclusion relation

$$\mathcal{H}_{\text{Gauss}}(\mathbb{S}^{d-1}) \subsetneq \mathcal{H}_{\text{Lap}}(\mathbb{S}^{d-1}) = \mathcal{H}_{N_1}(\mathbb{S}^{d-1}) \subseteq \mathcal{H}_{N_k}(\mathbb{S}^{d-1}),$$

where the four spaces denote the RKHS associated with the Gaussian kernel, Laplace kernel, the NTK of two-layer and $(k+1)$-layer ($k \geq 1$) fully connected neural networks, respectively. All four kernels are restricted to $\mathbb{S}^{d-1}$. However, the relation between $\mathcal{H}_{\text{Lap}}(\mathbb{S}^{d-1})$ and $\mathcal{H}_{N_k}(\mathbb{S}^{d-1})$ remains open in (Geifman et al., 2020).

We make a final conclusion on this problem and show that the RKHS of the Laplace kernel and the NTK with any number of layers have the same set of functions, when they are both restricted to $\mathbb{S}^{d-1}$. In other words, we prove the following theorem.

**Theorem 1.** *Let $\mathcal{H}_{\text{Lap}}(\mathbb{S}^{d-1})$ and $\mathcal{H}_{N_k}(\mathbb{S}^{d-1})$ be the RKHS associated with the Laplace kernel $K_{\text{Lap}}(x,y) = e^{-c\|x-y\|}$ ($c > 0$) and the neural tangent kernel of a $(k+1)$-layer fully connected*

*ReLU network. Both kernels are restricted to the sphere $\mathbb{S}^{d-1}$. Then the two spaces include the same set of functions:*

$$\mathcal{H}_{\mathrm{Lap}}(\mathbb{S}^{d-1}) = \mathcal{H}_{N_k}(\mathbb{S}^{d-1}), \quad \forall k \geq 1 \,.$$

Our second result is that the exponential power kernel with a smaller power (making the kernel less smooth) leads to a larger RKHS, both when it is restricted to the sphere $\mathbb{S}^{d-1}$ and when it is defined on the entire $\mathbb{R}^d$.

**Theorem 2.** *Let $\mathcal{H}_{K_{\exp}^{\gamma,\sigma}}(\mathbb{S}^{d-1})$ and $\mathcal{H}_{K_{\exp}^{\gamma,\sigma}}(\mathbb{R}^d)$ be the RKHS associated with the exponential power kernel $K_{\exp}^{\gamma,\sigma}(x,y) = \exp\left(-\frac{\|x-y\|^\gamma}{\sigma}\right)$ ($\gamma, \sigma > 0$) when it is restricted to the unit sphere $\mathbb{S}^{d-1}$ and defined on the entire $\mathbb{R}^d$, respectively. Then we have the following RKHS inclusions:*

*(1) If $0 < \gamma_1 < \gamma_2 < 2$,*

$$\mathcal{H}_{K_{\exp}^{\gamma_2,\sigma_2}}(\mathbb{S}^{d-1}) \subseteq \mathcal{H}_{K_{\exp}^{\gamma_1,\sigma_1}}(\mathbb{S}^{d-1}) \,.$$

*(2) If $0 < \gamma_1 < \gamma_2 < 2$ are rational,*

$$\mathcal{H}_{K_{\exp}^{\gamma_2,\sigma_2}}(\mathbb{R}^d) \subseteq \mathcal{H}_{K_{\exp}^{\gamma_1,\sigma_1}}(\mathbb{R}^d) \,.$$

If it is restricted to the unit sphere, the RKHS of the exponential power kernel with $\gamma < 1$ is even larger than that of NTK. This result partially explains the observation in (Hui et al., 2019) that the best performance is attained by a highly non-smooth exponential power kernel with $\gamma < 1$. Geifman et al. (2020) applied the exponential power kernel and the NTK to classification and regression tasks on the UCI dataset and other large scale datasets. Their experiment results also showed that the exponential power kernel slightly outperforms the NTK.

## 1.1 FURTHER RELATED WORK

Minh et al. (2006) showed the complete spectrum of the polynomial and Gaussian kernels on $\mathbb{S}^{d-1}$. They also gave a recursive relation for the eigenvalues of the polynomial kernel on the hypercube $\{-1, 1\}^d$. Prior to the NTK (Jacot et al., 2018), Cho & Saul (2009) presented a pioneering study on kernel methods for neural networks. Bach (2017) studied the eigenvalues of positively homogeneous activation functions of the form $\sigma_\alpha(u) = \max\{u, 0\}^\alpha$ (e.g., the ReLU activation when $\alpha = 1$) in their Mercer decomposition with Gegenbauer polynomials. Using the results in (Bach, 2017), Bietti & Mairal (2019) analyzed the two-layer NTK and its RKHS in order to investigate the inductive bias in the NTK regime. They studied the Mercer decomposition of two-layer NTK with ReLU activation on $\mathbb{S}^{d-1}$ and characterized the corresponding RKHS by showing the asymptotic decay rate of the eigenvalues in the Mercer decomposition with Gegenbauer polynomials. In their derivation of a more concise expression of the ReLU NTK, they used the calculation of (Cho & Saul, 2009) on arc-cosine kernels of degree 0 and 1. Cao et al. (2019) improved the eigenvalue bound for the $k$-th eigenvalue derived in (Bietti & Mairal, 2019) when $d \gg k$. Geifman et al. (2020) used the results in (Bietti & Mairal, 2019) and considered the two-layer ReLU NTK with bias $\beta$ initialized with zero, rather than initialized with a normal distribution (Jacot et al., 2018). However, neither (Bietti & Mairal, 2019) nor (Geifman et al., 2020) went beyond two layers when they tried to characterize the RKHS of the ReLU NTK. This line of work (Bach, 2017; Bietti & Mairal, 2019; Geifman et al., 2020) is closely related to the Mercer decomposition with spherical harmonics. Interested readers are referred to (Atkinson & Han, 2012) for spherical harmonics on the unit sphere. The concurrent work (Bietti & Bach, 2021) analyzed the eigenvalues of the ReLU NTK.

Arora et al. (2019a) presented a dynamic programming algorithm that computes convolutional NTK with ReLU activation. Yang & Salman (2019) analyzed the spectra of the conjugate kernel (CK) and NTK on the boolean cube. Fan & Wang (2020) studied the spectrum of the gram matrix of training samples under the CK and NTK and showed that their eigenvalue distributions converge to a deterministic limit. The limit depends on the eigenvalue distribution of the training samples.

## 2 PRELIMINARIES

Let $\mathbb{C}$ denote the set of all complex numbers and write $\mathbf{i} \triangleq \sqrt{-1}$. For $z \in \mathbb{C}$, write $\Re z, \Im z, \arg z \in (-\pi, \pi]$ for its real part, imaginary part, and argument, respectively. Let $\mathbb{H}^+ \triangleq \{z \in \mathbb{C} \mid \Im z > 0\}$

denote the upper half-plane and $\mathbb{H}^- \triangleq \{z \in \mathbb{C} \mid \Im z < 0\}$ denote the lower half-plane. Write $B_z(r)$ for the open ball $\{w \in \mathbb{C} \mid |z - w| < r\}$ and $\bar{B}_z(r)$ for the closed ball $\{w \in \mathbb{C} \mid |z - w| \leq r\}$.

Suppose that $f(z)$ has a power series representation $f(z) = \sum_{n \geq 0} a_n z^n$ around 0. Denote $[z^n]f(z) \triangleq a_n$ to be the coefficient of the $n$-th order term.

For two sequences $\{a_n\}$ and $\{b_n\}$, write $a_n \sim b_n$ if $\lim_{n \to \infty} \frac{a_n}{b_n} = 1$. Similarly, for two functions $f(z)$ and $g(z)$, write $f(z) \sim g(z)$ as $z \to z_0$ if $\lim_{z \to z_0} \frac{f(z)}{g(z)} = 1$. We also use big-$O$ and little-$o$ notation to characterize asymptotics.

Write $\mathscr{L}\{f(t)\}(s) \triangleq \int_0^\infty f(t)e^{-st}dt$ for the Laplace transform of a function $f(t)$. The inverse Laplace transform of $F(s)$ is denoted by $\mathscr{L}^{-1}\{F(s)\}(t)$.

## 2.1 POSITIVE DEFINITE KERNELS

For any positive definite kernel function $K(x, y)$ defined for $x, y \in E$, denote $\mathcal{H}_K(E)$ its associated reproducing kernel Hilbert space (RKHS). For any two positive definite kernel functions $K_1$ and $K_2$, we write $K_1 \preccurlyeq K_2$ if $K_2 - K_1$ is a positive definite kernel. For a complete review of results on kernels and RKHS, please see (Saitoh & Sawano, 2016).

We will study positive definite zonal kernels on the sphere $\mathbb{S}^{d-1} = \{x \in \mathbb{R}^d \mid \|x\| = 1\}$. For a zonal kernel $K(x, y)$, there exists a real function $\tilde{K} : [-1, 1] \to \mathbb{R}$ such that $K(x, y) = \tilde{K}(u)$, where $u = x^\top y$. We abuse the notation and use $K(u)$ to denote $\tilde{K}(u)$, i.e., $K(u)$ here is real function on $[-1, 1]$.

In the sequel, we introduce two instances of the positive definite kernel that this paper will investigate.

**Laplace Kernel**  The Laplace kernel $K_{\text{Lap}}(x, y) = e^{-c\|x-y\|}$ with $c > 0$ restricted to the sphere $\mathbb{S}^{d-1}$ is given by $K_{\text{Lap}}(x, y) = e^{-c\sqrt{2(1-x^\top y)}} = e^{-\tilde{c}\sqrt{1-u}} \triangleq K_{\text{Lap}}(u)$, where by our convention $u = x^\top y$ and $\tilde{c} \triangleq \sqrt{2}c > 0$. We denote its associated RKHS by $\mathcal{H}_{\text{Lap}}$.

**Exponential Power Kernel**  The exponential power kernel (Hui et al., 2019) with $\gamma > 0$ and $\sigma > 0$ is given by $K_{\exp}^{\gamma;\sigma}(x, y) = \exp\left(-\frac{\|x-y\|^\gamma}{\sigma}\right)$. If $x$ and $y$ are restricted to the sphere $\mathbb{S}^{d-1}$, we have $K_{\exp}^{\gamma;\sigma}(x, y) = \exp\left(-\frac{(2(1-x^\top y))^{\gamma/2}}{\sigma}\right)$.

**Neural Tangent Kernel**  Given the input $x \in \mathbb{R}^d$ (we define $d_0 \triangleq d$) and parameter $\theta$, this paper considers the following network model with $(k + 1)$ layers

$$
\begin{aligned}
&f_\theta(x) \\
&= w^\top \sqrt{\frac{2}{d_k}} \sigma \left( W_k \sqrt{\frac{2}{d_{k-1}}} \sigma \left( \cdots \sqrt{\frac{2}{d_2}} \sigma \left( W_2 \sqrt{\frac{2}{d_1}} \sigma \left( W_1 x + \beta b_1 \right) + \beta b_2 \right) \cdots \right) + \beta b_k \right) + \beta b_{k+1},
\end{aligned}
\tag{1}
$$

where the parameter $\theta$ encodes $W_l \in \mathbb{R}^{d_l \times d_{l-1}}$, $b_l \in \mathbb{R}^{d_l}$ $(l = 1, \ldots, k)$, $w \in \mathbb{R}^{d_k}$, and $b_{k+1} \in \mathbb{R}$. The weight matrices $W_1, \ldots, W_k, w$ are initialized with $\mathcal{N}(0, I)$ and the biases $b_1, \ldots, b_{k+1}$ are initialized with zero, where $\mathcal{N}(0, I)$ is the multivariate standard normal distribution. The activation function is chosen to be the ReLU function $\sigma(x) \triangleq \max\{x, 0\}$.

Geifman et al. (2020) and Bietti & Mairal (2019) presented the following recursive relations of the NTK $N_k(x, y)$ of the above ReLU network (1):

$$\Sigma_k(x, y) = \sqrt{\Sigma_{k-1}(x, x)\Sigma_{k-1}(y, y)} \kappa_1 \left( \frac{\Sigma_{k-1}(x, y)}{\sqrt{\Sigma_{k-1}(x, x)\Sigma_{k-1}(y, y)}} \right)$$

$$N_k(x, y) = \Sigma_k(x, y) + N_{k-1}(x, y) \kappa_0 \left( \frac{\Sigma_{k-1}(x, y)}{\sqrt{\Sigma_{k-1}(x, x)\Sigma_{k-1}(y, y)}} \right) + \beta^2 \,, \tag{2}$$

where $\kappa_0$ and $\kappa_1$ are the arc-cosine kernels of degree 0 and 1 (Cho & Saul, 2009) given by

$$\kappa_0(u) = \frac{1}{\pi}(\pi - \arccos(u)), \quad \kappa_1(u) = \frac{1}{\pi} \left( u \cdot (\pi - \arccos(u)) + \sqrt{1 - u^2} \right) \,.$$

The initial conditions are

$$N_0(x, y) = u + \beta^2, \quad \Sigma_0(x, y) = u \,, \tag{3}$$

where $u = x^\top y$ by our convention.

The NTKs defined in (Bietti & Mairal, 2019) and (Geifman et al., 2020) are slightly different. There is no bias term $\beta^2$ in (Bietti & Mairal, 2019), while the bias term appears in (Geifman et al., 2020). We adopt the more general setup with the bias term.

**Lemma 3** (Proof in Appendix A.1). $\Sigma_k(x, x) = 1$ *for any* $x \in \mathbb{S}^{d-1}$ *and* $k \geq 0$.

Lemma 3 simplifies (2) and gives

$$\Sigma_k(u) = \kappa_1^{(k)}(u) \,, \qquad N_k(u) = \kappa_1^{(k)}(u) + N_{k-1}(u)\kappa_0(\kappa_1^{(k-1)}(u)) + \beta^2 \,, \tag{4}$$

where $\kappa_1^{(k)}(u) \triangleq \underbrace{\kappa_1(\kappa_1(\cdots\kappa_1(\kappa_1(u))\cdots))}_{k}$ is the $k$-th iterate of $\kappa_1(u)$. For example, $\kappa_1^{(0)}(u) = u$, $\kappa_1^{(1)}(u) = \kappa_1(u)$ and $\kappa_1^{(2)}(u) = \kappa_1(\kappa_1(u))$. We present a detailed derivation of (4) in Appendix A.2.

## 3 RESULTS ON NEURAL TANGENT KERNEL

In this section, we present an overview of our proof for Theorem 1. Since (Geifman et al., 2020) showed $\mathcal{H}_{\mathrm{Lap}}(\mathbb{S}^{d-1}) \subseteq \mathcal{H}_{N_k}(\mathbb{S}^{d-1})$, it suffices to prove the reverse inclusion $\mathcal{H}_{N_k}(\mathbb{S}^{d-1}) \subseteq \mathcal{H}_{\mathrm{Lap}}(\mathbb{S}^{d-1})$. We then relate positive definite kernels with their RKHS according to the following lemma.

**Lemma 4** ((Aronszajn, 1950, p. 354) and (Saitoh & Sawano, 2016, Theorem 2.17)). *Let* $K_1, K_2 : \Omega \times \Omega \to \mathbb{C}$ *be two positive definite kernels. Then the Hilbert space* $\mathcal{H}_{K_1}$ *is a subset of* $\mathcal{H}_{K_2}$ *if and only if there exists some constant* $\gamma > 0$ *such that*

$$K_1 \preccurlyeq \gamma^2 K_2 \,.$$

Lemma 4 implies that in order to show $\mathcal{H}_{N_k}(\mathbb{S}^{d-1}) \subseteq \mathcal{H}_{\mathrm{Lap}}(\mathbb{S}^{d-1})$, it suffices to show $\gamma^2 K_{\mathrm{Lap}} - N_k$ is a positive definite kernel for some $\gamma > 0$. Note that both $K_{\mathrm{Lap}}$ and $N_k$ are positive definite kernels on the unit sphere. Then the Maclaurin series of $K_{\mathrm{Lap}}(u)$ and $N_k(u)$ have all non-negative coefficients by the classical approximation theory; see (Schoenberg, 1942, Theorem 2), Bingham (1973), and (Cheney & Light, 2009, Chapter 17). Conversely, if the Maclaurin series of $K(u)$ have all non-negative coefficients, $K(x, y) = K(x^\top y)$ is a positive definite kernel on the unit sphere. To be precise, we have the following lemma.

**Lemma 5** (Schoenberg (1942); Bingham (1973)). *Suppose that* $K(x, y) = f(x^\top y)$ *where* $x, y \in \mathbb{S}^{d-1}$ *and* $f$ *is continuous on* $[-1, 1]$.[1] *Then* $K$ *is a positive definite kernel on* $\mathbb{S}^{d-1}$ *for every* $d$ *if and only if* $f(u) = \sum_{k=0}^{\infty} a_k u^k$, *in which* $a_k \geq 0$ *and* $\sum_{k=0}^{\infty} a_k < \infty$.

Thus, we turn to show that there exists $\gamma > 0$ such that $\gamma^2 [z^n] K_{\mathrm{Lap}}(z) \geq [z^n] N_k(z)$ holds for every $n \geq 0$.

---

[1] When $x$ and $y$ live on the unit sphere (i.e., $x^\top x = y^\top y = 1$), their inner product $x^\top y$ can be any real number in $[-1, 1]$.

Exact calculation of the asymptotic rate of the Maclaurin coefficients is intractable for $N_k$ due to its recursive definition. Instead, we apply singularity analysis tools in analytic combinatorics. We refer the readers to (Flajolet & Sedgewick, 2009) for a systematic introduction. We treat all (zonal) kernels, $K_{\mathrm{Lap}}(u)$, $N_k(u)$, $\kappa_0(u)$, and $\kappa_1(u)$, as complex functions of variable $u \in \mathbb{C}$. To emphasize, we use $z \in \mathbb{C}$ instead of $u$ to denote the variable. The theory of analytic combinatorics states that the asymptotic of the coefficients of the Maclaurin series is determined by the local nature of the complex function at its dominant singularities (i.e., the singularities closest to $z = 0$).

To apply the methodology from (Flajolet & Sedgewick, 2009), we introduce some additional definitions. For $R > 1$ and $\phi \in (0, \pi/2)$, the $\Delta$-*domain* $\Delta(\phi, R)$ is defined by

$$\Delta(\phi, R) \triangleq \{ z \in \mathbb{C} \mid |z| < R, z \neq 1, |\arg(z - 1)| > \phi \}.$$

For a complex number $\zeta \neq 0$, a $\Delta$-domain at $\zeta$ is the image by the mapping $z \mapsto \zeta z$ of $\Delta(\phi, R)$ for some $R > 1$ and $\phi \in (0, \pi/2)$. A function is $\Delta$-analytic at $\zeta$ if it is analytic on a $\Delta$-domain at $\zeta$.

Suppose the function $f(z)$ has only one dominant singularity and without loss of generality assume that it lies at $z = 1$. We then have the following lemma.

**Lemma 6** ((Flajolet & Sedgewick, 2009, Corollary VI.1)). *If $f$ is $\Delta$-analytic at its dominant singularity* 1 *and*

$$f(z) \sim (1 - z)^{-\alpha}, \quad \text{as } z \to 1, z \in \Delta$$

*with $\alpha \notin \{0, -1, -2, \dots\}$, we have*

$$[z^n] f(z) \sim \frac{n^{\alpha - 1}}{\Gamma(\alpha)}.$$

If the function has multiple dominant singularities, the influence of each singularity is added up (See (Flajolet & Sedgewick, 2009, Theorem VI.5) for more details). Careful singularity analysis then gives

$$[z^n] K_{\mathrm{Lap}}(z) \sim C_1 n^{-3/2}, \quad [z^n] N_k(z) \leq C_2 n^{-3/2},$$

for some positive constants $C_1, C_2 > 0$. We refer to Section 3.2 and Appendix A.4 for more detailed steps. They are indeed of the same order of decay rate $n^{-3/2}$, which implies that such $\gamma$ exists. This shows $\mathcal{H}_{N_k}(\mathbb{S}^{d-1}) \subseteq \mathcal{H}_{\mathrm{Lap}}(\mathbb{S}^{d-1})$.

### 3.1 $\Delta$-Analyticity of Neural Tangent Kernels

We present the $\Delta$-analyticity of the NTKs here. In light of (4), the NTKs $N_k$ are compositions of arc-cosine kernels $\kappa_0$ and $\kappa_1$. We analytically extend $\kappa_0$ and $\kappa_1$ to a complex function of a complex variable $z \in \mathbb{C}$. Both complex functions $\arccos(z)$ and $\sqrt{1 - z^2}$ have branch points at $z = \pm 1$. Therefore, the branch cut of $\kappa_0(z)$ and $\kappa_1(z)$ is $[1, \infty) \cup (-\infty, -1]$. They have a single-valued analytic branch on

$$D = \mathbb{C} \setminus [1, \infty) \setminus (-\infty, -1]. \tag{5}$$

On this branch, we have

$$\kappa_0(z) = \frac{\pi + \mathbf{i} \log(z + \mathbf{i}\sqrt{1 - z^2})}{\pi},$$

$$\kappa_1(z) = \frac{1}{\pi} \left[ z \cdot \left( \pi + \mathbf{i} \log(z + \mathbf{i}\sqrt{1 - z^2}) \right) + \sqrt{1 - z^2} \right],$$

where we use the principal value of the logarithm and square root. We then show the dominant singularities of $\kappa_1^{(k)}(z)$ are $\pm 1$ and that $\kappa_1^{(k)}(z)$ is $\Delta$-analytic at $\pm 1$ for any $k \geq 1$. We further have the following theorem on the $\Delta$-singularity for $N_k$.

**Theorem 7** (Proof in Appendix A.3). *For each $k \geq 1$, the dominant singularities of $N_k$ are $\pm 1$. There exists $R_k > 1$ such that $N_k$ is analytic on $\{z \in \mathbb{C} \mid |z| \leq R_k\} \cap D$, where $D = \mathbb{C} \setminus [1, \infty) \setminus (-\infty, -1]$.*

## 3.2 Asymptotic Rates of Maclaurin Coefficients for $N_k$

The following theorem demonstrates the asymptotic rates of Maclaurin coefficients for $N_k$.

**Theorem 8** (Proof in Appendix A.4). *The $n$-th order coefficient of the Maclaurin series of the $(k+1)$-layer NTK in (2) satisfies $[z^n]N_k(z) = O(n^{-3/2})$.*

In the proof of Theorem 8, we show the following asymptotics

$$N_k(z) = (k+1)(z+\beta^2) - \left( \sqrt{2}(1+\beta^2)\frac{k(k+1)}{2\pi} + o(1) \right) \sqrt{1-z} \quad \text{as } z \to 1, \tag{6}$$

$$N_k(z) = N_k(-1) + \left( \frac{\sqrt{2}(\beta^2-1)}{\pi} \prod_{j=1}^{k-1} \kappa_0(\kappa_1^j(-1)) + o(1) \right) \sqrt{1+z} \quad \text{as } z \to -1. \tag{7}$$

When $\beta = 1$, the singularity at $z = -1$ will not provide a $\sqrt{1+z}$ term. The dominating term in (7) is a higher power of $\sqrt{1+z}$. As a result, the contribution of the singularity at $-1$ to the Maclaurin coefficients is $o(n^{-3/2})$ and dominated by the contribution of the singularity at 1. The singularity at $z = 1$ provides a $\sqrt{1-z}$ term and thus contributes to $O(n^{-3/2})$ decay rate of $[z^n]N_k(z)$. In addition, from (6), we deduce

$$\frac{[z^n]N_k(z)}{n^{-3/2}} \sim -\frac{2\sqrt{2}k(k+1)}{(2\pi)\Gamma\left(-\frac{1}{2}\right)} = \frac{k(k+1)}{\sqrt{2}\pi^{3/2}}. \tag{8}$$

When $\beta \neq 1$, both singularities $\pm 1$ contribute $\Theta(n^{-3/2})$ to the Maclaurin cofficients. The contribution of $z = 1$ is

$$-\frac{\sqrt{2}(1+\beta^2)k(k+1)}{2\pi\Gamma\left(-\frac{1}{2}\right)}n^{-3/2} = \frac{(\beta^2+1)k(k+1)}{2\sqrt{2}\pi^{3/2}}n^{-3/2}.$$

The contribution of $z = -1$ is

$$\left( \frac{\sqrt{2}(\beta^2-1)}{\pi\Gamma(-1/2)} \prod_{j=1}^{k-1} \kappa_0(\kappa_1^j(-1)) \right) n^{-3/2} = \left( \frac{1-\beta^2}{\sqrt{2}\pi^{3/2}} \prod_{j=1}^{k-1} \kappa_0(\kappa_1^j(-1)) \right) n^{-3/2}.$$

Combining them gives

$$\frac{[z^n]N_k(z)}{n^{-3/2}} \sim \frac{(\beta^2+1)k(k+1)}{2\sqrt{2}\pi^{3/2}} + (-1)^n \frac{1-\beta^2}{\sqrt{2}\pi^{3/2}} \prod_{j=1}^{k-1} \kappa_0(\kappa_1^j(-1)). \tag{9}$$

Based on Theorem 8, we are ready to prove Theorem 1.

*Proof.* Let $K_{\text{Lap}}(z) = e^{-c\sqrt{1-z}}$, where $c > 0$ is an arbitrary constant. We have $\mathcal{H}_{K_{\text{Lap}}} = \mathcal{H}_{\text{Lap}}$. The complex function $K_{\text{Lap}}$ is analytic on $\mathbb{C} \setminus [1, \infty)$. As $z \to 1$, we have

$$\frac{K_{\text{Lap}}(z) - 1}{-c} = \sqrt{1-z} + o(\sqrt{1-z}) \sim \sqrt{1-z}.$$

By Lemma 6, we obtain

$$[z^n]K_{\text{Lap}}(z) \sim \frac{c}{2\sqrt{\pi}}n^{-3/2}. \tag{10}$$

Note that $[z^n]N_k(z) = O(n^{-3/2})$ from Theorem 8. Therefore, there exists $\gamma > 0$ such that $\gamma^2 \cdot [z^n]K_{\text{Lap}}(z) - [z^n]N_k(z) > 0$ for all $n \geq 0$. This further implies $\gamma^2 K_{\text{Lap}}(x^\top y) - N_k(x^\top y)$ is a positive definite kernel. According to Lemma 4, we have $\mathcal{H}_{N_k}(\mathbb{S}^{d-1}) \subseteq \mathcal{H}_{\text{Lap}}(\mathbb{S}^{d-1})$. Note that, due to (Geifman et al., 2020, Theorem 3), we also have $\mathcal{H}_{\text{Lap}}(\mathbb{S}^{d-1}) \subseteq \mathcal{H}_{N_k}(\mathbb{S}^{d-1})$. Therefore, for any $k \geq 1$, $\mathcal{H}_{\text{Lap}}(\mathbb{S}^{d-1}) = \mathcal{H}_{N_k}(\mathbb{S}^{d-1})$.

$\square$

## 4   RESULTS ON EXPONENTIAL POWER KERNEL

This section presents the proof of Theorem 2. We first show part (1) below by singularity analysis.

*Proof of part (1) of Theorem 2.* Recall that the exponential power kernel restricted to the unit sphere with $\gamma > 0$ and $\sigma > 0$ is given by $K_{\exp}^{\gamma,\sigma}(x,y) = \exp\left(-\frac{\|x-y\|^\gamma}{\sigma}\right) = \exp\left(-\frac{(2(1-x^\top y))^{\gamma/2}}{\sigma}\right)$. Let us study the decay rate of the Maclaurin coefficients of $K_{\exp}^{\gamma,\sigma}(z) \triangleq e^{-c(1-z)^{\gamma/2}}$, where $c = 2^{\gamma/2}/\sigma$. The dominant singularity lies at $z = 1$. As $z \to 1$, we get

$$K_{\exp}^{\gamma,\sigma}(z) = 1 - (c + o(1))(1-z)^{\gamma/2}.$$

Applying Lemma 6 gives $[z^n]K_{\exp}^{\gamma,\sigma}(z) \sim \frac{cn^{-\gamma/2-1}}{-\Gamma(-\gamma/2)}$. Therefore, a smaller $\gamma$ results in a larger RKHS. □

Part (2) of Theorem 2 requires more technical preparation. Recall that $\mathscr{L}$ and $\mathscr{L}^{-1}$ denote the Laplace transform and inverse Laplace transform, respectively. We explicitly calculate the inverse Laplace transform $\mathscr{L}^{-1}\{\exp(-s^a)\}(t)$ using Bromwich contour integral and get the following lemma.

**Lemma 9** (Proof in Appendix B.1)**.** *For $a \in (0,1)$, $f(t) \triangleq \mathscr{L}^{-1}\{\exp(-s^a)\}(t)$ exists. Moreover, $f(t)$ is continuous in $-\infty < t < \infty$ and satisfies $f(0) = 0$. If $t > 0$, we have*

$$f(t) = \frac{1}{\pi}\sum_{k=0}^{\infty} \frac{(-1)^{k+1}\Gamma(ak+1)\sin(\pi ak)}{k!t^{ak+1}}. \tag{11}$$

Based on the series representation (11), we then analyze the asymptotic rate for $f(t)$ when $a$ is rational. Note that if $a \in (0,1)$, we have $-\frac{1}{\Gamma(-a)} > 0$.

**Lemma 10** (Proof in Appendix B.2)**.** *Let $f(t)$ be as defined in Lemma 9. For $a = \frac{p}{q} \in (0,1)$ ($p$ and $q$ are co-prime), we have $f(t) \sim -\frac{1}{t^{a+1}\Gamma(-a)}$ as $t \to +\infty$.*

Thus, We have the following corollary for general exponential power kernel.

**Corollary 11.** *For $a = \frac{p}{q} \in (0,1)$ ($p$ and $q$ are co-prime) and $\sigma > 0$, $\mathscr{L}^{-1}\{\exp(-s^a/\sigma)\}(t)$ is continuous in $t \in \mathbb{R}$ and satisfies $\mathscr{L}^{-1}\{\exp(-s^a/\sigma)\}(0) = 0$. Moreover, $\mathscr{L}^{-1}\{\exp(-s^a/\sigma)\}(t) \sim Ct^{-a-1}$ as $t \to +\infty$, for some constant $C > 0$.*

*Proof.* Use the property $\mathscr{L}^{-1}\{F(cs)\}(t) = \frac{1}{c}f\left(\frac{t}{c}\right)$, where $c > 0$ and $F(s) = \mathscr{L}\{f(t)\}(s)$. □

Before completing the proof for part (2), we need two additional lemmas from the classical approximation theory. Recall that a function $f(t)$ is *completely monotone* if it is continuous on $[0,\infty)$, infinitely differentiable on $(0,\infty)$ and satisfies $(-1)^n\frac{d^n f(t)}{dt} \geq 0$ for every $n = 0,1,2,\ldots$ and $t > 0$ (Cheney & Light, 2009, Chapter 14).

**Lemma 12** (Schoenberg interpolation theorem (Cheney & Light, 2009, Theorem 1 of Chapter 15))**.** *If $f$ is completely monotone but not constant on $[0,\infty)$, then for any $n$ distinct points $x_1, x_2, \ldots, x_n$ in any inner-product space, the matrix $A_{ij} = f(\|x_i - x_j\|^2)$ is positive definite.*

**Lemma 13** (Bernstein-Widder (Cheney & Light, 2009, Theorem 1 of Chapter 14))**.** *A function $f : [0,\infty) \to [0,\infty)$ is completely monotone if and only if there is a nondecreasing bounded function $g$ such that $f(t) = \int_0^\infty e^{-st}dg(s)$.*

Now we are ready to prove part (2).

*Proof of part (2) of Theorem 2.* By Lemma 12 and Lemma 4, we need to show that

$$c^2\exp(-x^{\gamma_1/2}/\sigma_1) - \exp(-x^{\gamma_2/2}/\sigma_2) \tag{12}$$

is completely monotone but not constant on $[0, \infty)$ for some $c > 0$. By Lemma 13, it suffices to check that (12) is the Laplace transform of a non-negative function on $[0, \infty)$. By Corollary 11, for rational $\gamma_1, \gamma_2 \in (0, 1]$, there exists $c > 0$ such that

$$c^2 \mathscr{L}^{-1}\{\exp(-x^{\gamma_1/2}/\sigma_1)\} - \mathscr{L}^{-1}\{\exp(-x^{\gamma_2/2}/\sigma_2)\}$$

is continuous and positive on $[0, \infty)$, which completes the proof. $\qquad\square$

## 5 NUMERICAL RESULTS

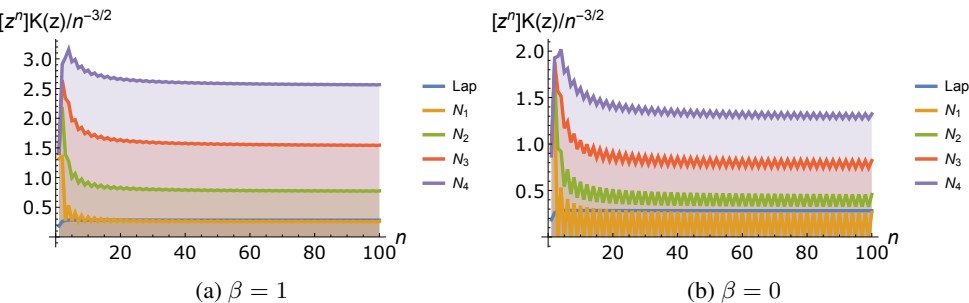

Figure 1: We plot $[z^n]K(z)/n^{-3/2}$ versus $n$ for the Laplace kernel $K_{\mathrm{Lap}}(u) = e^{-\sqrt{2(1-u)}}$ and NTKs $N_1, \ldots, N_4$ with $\beta = 0, 1$.

| Kernel $K$ | $\dfrac{[z^{100}]K(z)}{100^{-3/2}}$ $(\beta = 1)$ | Theory $(\beta = 1)$ | $\dfrac{[z^{100}]K(z)}{100^{-3/2}}$ $(\beta = 0)$ | Theory $(\beta = 0)$ |
|---|---|---|---|---|
| $K_{\mathrm{Lap}}$ | 0.28244 | $\frac{1}{2\sqrt{\pi}} \approx 0.282095$ | | |
| $N_1$ | 0.261069 | $\frac{\sqrt{2}}{\pi^{3/2}} \approx 0.253975$ | 0.261069 | $\frac{\sqrt{2}}{\pi^{3/2}} \approx 0.253975$ |
| $N_2$ | 0.776014 | $\frac{3\sqrt{2}}{\pi^{3/2}} \approx 0.761924$ | 0.457426 | $\frac{7}{2\sqrt{2}\pi^{3/2}} \approx 0.444455$ |
| $N_3$ | 1.54607 | $\frac{6\sqrt{2}}{\pi^{3/2}} \approx 1.52385$ | 0.821694 | $\frac{13\pi - \arccos(\pi^{-1})}{2\sqrt{2}\pi^{5/2}} \approx 0.800218$ |
| $N_4$ | 2.56559 | $\frac{10\sqrt{2}}{\pi^{3/2}} \approx 2.53975$ | 1.32472 | Equation (13) $\approx 1.29531$ |

Table 1: We report the numerical values of $\frac{[z^{100}]K(z)}{100^{-3/2}}$ for the Laplace kernel $K_{\mathrm{Lap}}(u) = e^{-\sqrt{2(1-u)}}$ and NTKs $N_1, \ldots, N_4$ with $\beta = 0, 1$. These numerical values are the final values of the curves in Fig. 1. We present the theoretical prediction by the asymptotic of $[z^n]K(z)/n^{-3/2}$ alongside each numerical value. The choice of $\beta$ does not apply to the Laplace kernel. Therefore, we only show the results of the Laplace kernel in the columns for $\beta = 1$ and leave blank the columns for $\beta = 0$.

We verify the asymptotics of the Maclaurin coefficients of the Laplace kernel and NTKs through numerical results.

Fig. 1 plots $\frac{[z^n]K(z)}{n^{-3/2}}$ versus $n$ for different kernels, including the Laplace kernel $K_{\mathrm{Lap}}(u) = e^{-\sqrt{2(1-u)}}$ and NTKs $N_1, \ldots, N_4$ with $\beta = 0, 1$. All curves converge to a constant as $n \to \infty$, which indicates that for every kernel $K(z)$ considered here, we have $[z^n]K(z) = \Theta(n^{-3/2})$. The numerical results agree with our theory in the proofs of Theorem 8 and Theorem 1.

Now we investigate the value of $[z^n]K(z)/n^{-3/2}$. Table 1 reports $[z^{100}]K(z)/100^{-3/2}$ for the Laplace kernel and NTKs with $\beta = 0, 1$. These numerical values are the final values of the curves in Fig. 1. The theoretical predictions are obtained through the asymptotic of $[z^n]K(z)/n^{-3/2}$, which we shall explain below. The theoretical prediction of $[z^{100}]N_4(z)/100^{-3/2}$ with $\beta = 0$ is presented below due to the space limit in the table

$$\frac{20 + \pi^{-2}\left(\pi - \arccos\left(\pi^{-1}\right)\right)\left(\pi - \arccos\left(\frac{\sqrt{\pi^2-1}+\pi-\arccos(\pi^{-1})}{\pi^2}\right)\right)}{2\sqrt{2}\pi^{3/2}} \approx 1.29531. \qquad (13)$$

We observe that the theoretical prediction by the asymptotic is close to the corresponding numerical value. There are two possible reasons that account for the minor discrepancy between them. First, the theoretical prediction reflects the situation for an infinitely large $n$ (so that the lower order terms become negligible), while $n = 100$ is clearly finite. Second, the numerical results for the Maclaurin series are obtained by numerical Taylor expansion and therefore numerical errors could be present.

In what follows, we explain how to obtain the theoretical predictions. First, (10) gives $[z^n]K_{\mathrm{Lap}}(z)/n^{-3/2} \sim \frac{1}{2\sqrt{\pi}}$. As a result, the theoretical prediction for $[z^{100}]K_{\mathrm{Lap}}(z)/100^{-3/2}$ is $\frac{1}{2\sqrt{\pi}}$. Now we explain the thereotical predictions for NTKs. When $\beta = 1$, the theoretical prediction is given by (8). We present it in the third column of Table 1 for $N_1, \ldots, N_4$. When $\beta = 0$, we plug $\beta = 0$ into (9) and obtain $\frac{[z^n]N_k(z)}{n^{-3/2}} \sim \frac{k(k+1)}{2\sqrt{2}\pi^{3/2}} + \frac{(-1)^n}{\sqrt{2}\pi^{3/2}} \prod_{j=1}^{k-1} \kappa_0(\kappa_1^j(-1))$. The above expression (when $n = 100$ on the right-hand side) is the theoretical value presented in the fifth column of Table 1 for NTKs.

## 6 DISCUSSION

Our result provides further evidence that the NTK is similar to the existing Laplace kernel. However, the following mysteries remain open. First, if we still restrict them to the unit sphere, do they have a similar learning dynamic when we perform kernelized gradient descent? Second, what is the behavior of the NTK and the Laplace kernel outside of $\mathbb{S}^{d-1}$ and in the entire space $\mathbb{R}^d$? Do they still share similarities in terms of the associated RKHS? If not, how far do they deviate from each other and is the difference significant? Third, this work along with (Bietti & Mairal, 2019; Geifman et al., 2020) focuses on the NTK with ReLU activation. It would be interesting to explore the influence of different activations upon the RKHS and other kernel-related quantities. We would like to remark that the ReLU NTK has a clean expression partly because the expectation over the Gaussian process in the general NTK can be computed exactly if the activation function is ReLU (which may not be true for other non-linearities, for example, it may require more work for sigmoid). Fourth, we showed that highly non-smooth exponential power kernels have an even larger RKHS than the NTK. It would be worthwhile comparing the performance of these non-smooth kernels and deep neural networks through more extensive experiments in a variety of machine learning tasks.

Moreover, we show that a less smooth exponential power kernel leads to a larger RKHS and therefore greater expressive power. Its generalization capability is a related but different topic. Analyzing the generalization error requires more efforts in general. Researchers often use the RKHS norm to provide an upper bound for it. We will study its generalization in future work.

### ACKNOWLEDGEMENTS

We gratefully acknowledge the support of the Simons Institute for the Theory of Computing. We thank Peter Bartlett, Mikhail Belkin, Jason D. Lee, and Iosif Pinelis for helpful discussions and thank Mikhail Belkin and Alexandre Eremenko for introducing to us the works (Hui et al., 2019; Liu et al., 2020) and (Flajolet & Sedgewick, 2009), respectively.

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

# Appendices

## Table of Contents

## A   PROOFS FOR NEURAL TANGENT KERNEL

### A.1   PROOF OF LEMMA 3

*Proof.* We show it by induction. It holds when $k = 0$ by the initial condition (3). Assume that it holds for some $k \geq 0$, i.e., $\Sigma_k(x, x) = 1$. Consider $k + 1$. We have

$$\Sigma_{k+1}(x, x) = \kappa_1(\Sigma_k(x, x)) = \kappa_1(1) = 1 \,.$$

$\square$

### A.2   PROOF OF EQUATION (4)

*Proof.* We plug $\Sigma_k(x, x) = 1$ into (2) and obtain

$$\Sigma_k(x, y) = \kappa_1(\Sigma_{k-1}(x, y))$$
$$N_k(x, y) = \Sigma_k(x, y) + N_{k-1}(x, y)\kappa_0(\Sigma_{k-1}(x, y)) + \beta^2 \,.$$

Recall $\Sigma_0(x, y) = u$. By induction, we get

$$\Sigma_k(u) = \kappa_1^{(k)}(u) \,,$$

where $\kappa_1^{(k)}(u) \triangleq \kappa_1^{(k)}(u) = \underbrace{\kappa_1(\kappa_1(\cdots \kappa_1(\kappa_1(u))\cdots))}_{k}$ is the $k$-th iterate of $\kappa_1(u)$. Then it follows

$$N_k(u) = \kappa_1^{(k)}(u) + N_{k-1}(u)\kappa_0(\kappa_1^{k-1}(u)) + \beta^2 \,.$$

$\square$

### A.3   PROOF OF THEOREM 7

Lemma 14 and Lemma 15 demonstrate that $\pm 1$ are indeed singularities and analyze the asymptotics for $\kappa_1^{(k)}$ as $z$ tends to $\pm 1$, respectively. Our calculation is inspired by Pinelis (2020), which only considers $k = 2$.

**Lemma 14.** *For every $k \geq 1$, there exists $c_k(z)$ such that*

$$\kappa_1^{(k)}(z) = z + c_k(z)(1 - z)^{3/2} \,,$$

*where*

$$\lim_{z \to 1} c_k(z) = \frac{2\sqrt{2}k}{3\pi} \,.$$

*Proof.* We prove by induction on $k$. We first prove the statement for $k = 1$. Let $z = 1 - re^{i\theta}$. Taylor's theorem around 1 with integral form of remainder gives

$$\kappa_1(z) = z + \int_\gamma \frac{z - w}{\pi\sqrt{1 - w^2}} dw \,,$$

where $\gamma : [0, 1] \to \mathbb{C}$ is the simple straight line connecting 1 and $z$ taking the form $\gamma(t) = 1 - tre^{i\theta}$. It follows

$$\kappa_1(z) = z + \int_\gamma \frac{z - w}{\pi\sqrt{1 - w}} \cdot \frac{1}{\sqrt{1 + w}} dw$$

$$= z + \int_\gamma \frac{z - w}{\pi\sqrt{2}\sqrt{1 - w}} dw + \int_\gamma \frac{z - w}{\pi\sqrt{2}\sqrt{1 - w}} \cdot (\frac{\sqrt{2}}{\sqrt{1 + w}} - 1) dw \,.$$

Since

$$\int_\gamma \frac{z - w}{\sqrt{1 - w}} dw = \frac{2}{3}\sqrt{1 - w}(w - 3z + 2)\Big|_{w=1}^{w=z} = \frac{4}{3}(1 - z)^{3/2} \,,$$

we have

$$\kappa_1(z) = z + \frac{2\sqrt{2}}{3\pi}(1 - z)^{3/2} + \int_\gamma \frac{z - w}{\pi\sqrt{2}\sqrt{1 - w}} \cdot (\frac{\sqrt{2}}{\sqrt{1 + w}} - 1) dw \,.$$

We then turn to show

$$\lim_{z \to 1} \left\{ (1 - z)^{-3/2} \cdot \int_\gamma \frac{z - w}{\pi\sqrt{2}\sqrt{1 - w}} \cdot (\frac{\sqrt{2}}{\sqrt{1 + w}} - 1) dw \right\} = 0 \,.$$

Direct calculation gives

$$\lim_{z \to 1} \left\{ (1 - z)^{-3/2} \cdot \int_\gamma \frac{z - w}{\sqrt{1 - w}} \cdot (\frac{\sqrt{2}}{\sqrt{1 + w}} - 1) dw \right\}$$

$$= \lim_{r \to 0} \left\{ (re^{i\theta})^{-3/2} \cdot \int_0^1 \frac{(1 - t)r^2 e^{2i\theta}}{\sqrt{tre^{i\theta}}} (\frac{\sqrt{2}}{\sqrt{2 - tre^{i\theta}}} - 1) dt \right\}$$

$$= \lim_{r \to 0} \left\{ \int_0^1 \frac{1 - t}{\sqrt{t}} (\frac{1}{\sqrt{1 - tre^{i\theta}/2}} - 1) dt \right\}$$

$$= 0 \,.$$

Therefore, there exists $c_1(z)$ such that $\lim_{z \to 1} c_1(z) = \frac{2\sqrt{2}}{3\pi} \neq 0$ and

$$\kappa_1(z) = z + c_1(z)(1 - z)^{3/2} \,.$$

Next, assume that the desired equation holds for some $k \geq 1$. We then have

$$\kappa_1^{(k+1)}(z) = \kappa_1(\kappa_1^{(k)}(z))$$

$$= \kappa_1(z + c_k(z)(1 - z)^{3/2})$$

$$= z + c_k(z)(1 - z)^{3/2} + c_1\left(\kappa_1^{(k)}(z)\right) \cdot \left(1 - z - c_k(z)(1 - z)^{3/2}\right)^{3/2}$$

$$= z + c_{k+1}(z)(1 - z)^{3/2} \,,$$

where $c_{k+1}(z) \sim c_k(z) + c_1(k_1^{(k)}(z))$. Recall that when $z \to 1$, we have $\kappa_1^{(k)}(z) \to 1$ as well. Therefore we deduce

$$\lim_{z \to 1} c_{k+1}(z) = \lim_{z \to 1} c_k(z) + \lim_{z \to 1} c_1(k_1^k(z)) = \frac{2\sqrt{2}k}{3\pi} \neq 0 \,.$$

$\square$

**Lemma 15.** *For every $k \geq 1$, there exist $a_k \in \mathbb{R}$ and a complex function $b_k(z)$ such that*

$$\kappa_1^{(k)}(z) = a_k + b_k(z)(z+1)^{3/2},$$

*where*

$$a_k = \kappa_1^{(k)}(-1) \quad \text{and} \quad \lim_{z \to -1} b_k(z) = \frac{2\sqrt{2}}{3\pi} \prod_{j=1}^{k-1} \kappa_1'(\kappa_1^{(j)}(-1)) > 0.$$

*Proof.* We prove by induction on $k$. We first prove the statement for $k = 1$. Let $z = -1 + re^{i\theta}$. Taylor's theorem around $-1$ with integral form of remainder gives

$$\kappa_1(z) = \int_\gamma \frac{z - w}{\pi \sqrt{1 - w^2}} dw.$$

where $\gamma : [0,1] \to \mathbb{C}$ is the simple straight line connecting $-1$ and $z$ taking the form $\gamma(t) = -1 + tre^{i\theta}$. Similar arguments as in the proof of Lemma 14 give

$$\kappa_1(z) = b_1(z)(z+1)^{3/2},$$

where $\lim_{z \to -1} b_1(z) = \frac{2\sqrt{2}}{3\pi}$.

Next, assume that the desired equation holds for some $k \geq 1$. Define $h_k \triangleq \kappa_1^{(k)}(-1)$. Since $\kappa_1$ is strictly increasing on $[-1, 1]$, $\kappa_1(-1) = 0$ and $\kappa_1(1) = 1$, we have $h_1 = 0$ and $h_k \in (0, 1)$ for all $k > 1$. Expanding $\kappa_1$ around $h_k$ yields

$$\kappa_1(z) = \kappa_1(h_k) + p(z)(z - h_k) = h_{k+1} + p(z)(z - h_k),$$

where $\lim_{z \to h_k} p(z) = \kappa_1'(h_k)$. It follows that

$$\kappa_1^{k+1}(z) = \kappa_1(a_k + b_k(z)(z+1)^{3/2}) = h_{k+1} + p(\kappa_1^{(k)}(z))(a_k + b_k(z)(z+1)^{3/2} - h_k)$$

$$= a_{k+1} + b_{k+1}(z)(z+1)^{3/2},$$

where $a_{k+1} = h_{k+1} + \kappa_1'(h_k)(a_k - h_k)$ and $\lim_{z \to -1} b_{k+1}(z) = \kappa_1'(h_k) \lim_{z \to -1} b_k(z)$. By induction, we can show that $a_k = h_k$ for all $k \geq 1$. Since $\kappa_1'$ is strictly increasing on $[-1, 1]$, $\kappa_1'(-1) = 0$, and $\kappa_1'(1) = 1$, we have $\kappa_1'(h_k) \geq \kappa_1'(0) > 0$. As a result,

$$\lim_{z \to -1} b_{k+1}(z) = \frac{2\sqrt{2}}{3\pi} \prod_{j=1}^{k} \kappa_1'(\kappa_1^{(j)}(-1)) > 0.$$

$\square$

In the sequel, we show that $\pm 1$ are the only dominant singularities of $\kappa_1^{(k)}$ and $\kappa_1^{(k)}$ is $\Delta$-analytic at $\pm 1$ (Lemma 19).

**Lemma 16.** *For any $z \in \mathbb{C}$ with $\arg z \in (0, \pi/4)$, $\kappa_1(z) \in \mathbb{H}^+$. For any $z \in \mathbb{C}$ with $\arg z \in (-\pi/4, 0)$, $\kappa_1(z) \in \mathbb{H}^-$.*

*Proof.* The second part of the statement follows from the first according to the reflection principle. We only prove the first part here. Let $z = re^{i\theta}$ with $\theta \in (0, \pi/4)$. Taylor's theorem with integral form of the remainder and direct calculation give

$$\kappa_1(z) = \kappa_1(0) + \kappa_1'(0)z + \int_\gamma (z - w)\kappa_1''(w)dw = \frac{1}{\pi} + \frac{1}{2}z + \int_\gamma \frac{z - w}{\pi \sqrt{1 - w^2}} dw,$$

where $\gamma : [0, 1] \to \mathbb{C}$ is the simple straight line connecting $0$ and $z$ taking the form $\gamma(t) = tre^{i\theta}$. Then we have

$$\int_\gamma \frac{z - w}{\pi \sqrt{1 - w^2}} dw = r^2 e^{2i\theta} \int_0^1 \frac{1 - t}{\pi \sqrt{1 - r^2 t^2 e^{2i\theta}}} dt = e^{2i\theta} \int_0^r \frac{r - t}{\pi \sqrt{1 - t^2 e^{2i\theta}}} dt.$$

Since $\theta \in (0, \pi/4)$, we have $\arg(1 - t^2 e^{2i\theta}) \in (-\pi, 0)$. Further

$$\arg\left(\frac{1}{\sqrt{1 - t^2 e^{2i\theta}}}\right) \in (0, \pi/2) \qquad \text{and} \qquad \arg\left(\int_0^r \frac{r - t}{\pi\sqrt{1 - t^2 e^{2i\theta}}} dt\right) \in (0, \pi/2).$$

Noting $\arg(e^{2i\theta}) \in (0, \pi/2)$, we get

$$\arg\left(\int_\gamma \frac{z - w}{\pi\sqrt{1 - w^2}} dw\right) \in (0, \pi),$$

which gives a positive imaginary part. Combining with $\Im(1/\pi + z/2) > 0$ yields the desired statement. $\square$

**Lemma 17.** *For every $k \geq 1$ and $\varepsilon > 0$, there exists $\delta > 0$ such that $\kappa_1^{(k)}$ is analytic on $B_1(\delta) \cap \mathbb{H}^+$ and $B_1(\delta) \cap \mathbb{H}^-$ with*

$$\kappa_1^{(k)}(B_1(\delta) \cap \mathbb{H}^+) \subseteq B_1(\varepsilon) \cap \mathbb{H}^+,$$
$$\kappa_1^{(k)}(B_1(\delta) \cap \mathbb{H}^-) \subseteq B_1(\varepsilon) \cap \mathbb{H}^-.$$

*Proof.* We present the proof for $\mathbb{H}^+$ here and that for $\mathbb{H}^-$ can be shown similarly. We adopt an induction argument on $k$.

For $k = 1$, $\kappa_1$ is analytic on $\mathbb{H}^+$. Since $\kappa_1$ is continuous at $z = 1$, for any $\varepsilon > 0$, there exists $0 < \delta < 1/2$ such that

$$\kappa_1(B_1(\delta) \cap \mathbb{H}^+) \subseteq B_1(\varepsilon).$$

Lemma 16 implies $\kappa_1(B_1(\delta) \cap \mathbb{H}^+) \subseteq \mathbb{H}^+$. Combining them yields

$$\kappa_1(B_1(\delta) \cap \mathbb{H}^+) \subseteq B_1(\varepsilon) \cap \mathbb{H}^+. \tag{14}$$

Now assume that the statement holds true for some $k \geq 1$. Note that for any $\varepsilon > 0$, there exists $0 < \delta < 1/2$ such that (14) holds. Then by induction hypothesis, for this chosen $\delta$, there exists $\delta_1 > 0$ such that $\kappa_1^{(k)}$ is analytic on $B_1(\delta_1) \cap \mathbb{H}^+$ and

$$\kappa_1^{(k)}(B_1(\delta_1) \cap \mathbb{H}^+) \subseteq B_1(\delta) \cap \mathbb{H}^+.$$

It follows

$$\kappa_1^{(k+1)}(B_1(\delta_1) \cap \mathbb{H}^+) \subseteq \kappa_1(B_1(\delta) \cap \mathbb{H}^+) \subseteq B_1(\varepsilon) \cap \mathbb{H}^+.$$

This completes the proof. $\square$

**Lemma 18.** $|\kappa_1(z)| \leq 1$ *for any $|z| \leq 1$, where the equality holds if and only if $z = 1$.*

*Proof.* The Taylor series of $\kappa_1$ around $z = 0$ is

$$\kappa_1(z) = \frac{1}{\pi} + \frac{z}{2} + \sum_{n=1}^\infty \frac{(2n - 3)!!}{(2n - 1)n!2^n\pi} z^{2n}.$$

Therefore, for $|z| \leq 1$, we have

$$|\kappa_1(z)| \leq \frac{1}{\pi} + \frac{|z|}{2} + \sum_{n=1}^\infty \frac{(2n - 3)!!}{(2n - 1)n!2^n\pi} |z|^{2n} \leq \kappa_1(1) = 1.$$

The equality holds if and only if $z = 1$. $\square$

**Lemma 19.** *For each $k \geq 1$, there exists $R > 1$ such that $\kappa_1^{(k)}$ is analytic on $\{z \in \mathbb{C} \mid |z| \leq R\} \cap D$, where $D = \mathbb{C} \setminus [1, \infty) \setminus (-\infty, -1]$.*

*Proof.* For any $0 < \theta < \pi/2$, there exists $\delta_\theta > 0$ such that for all $|z| \leq 1$ with $|\arg z| \geq \theta$, we have

$$|\kappa_1(z)| \leq 1 - \delta_\theta \,.$$

To see this, we use an argument similar to (Pinelis, 2020). If we define $\phi \triangleq \arg z$, we have

$$\left| \frac{1}{\pi} + \frac{z}{2} \right| = \sqrt{\frac{|z|^2}{4} + \frac{|z| \cos \phi}{\pi} + \frac{1}{\pi^2}}$$

$$\leq \sqrt{\frac{1}{4} + \frac{\cos \theta}{\pi} + \frac{1}{\pi^2}} = \sqrt{\left( \frac{1}{2} + \frac{1}{\pi} \right)^2 - \frac{1 - \cos \theta}{\pi}} = \frac{1}{2} + \frac{1}{\pi} - \delta_\theta \,,$$

for some $\delta_\theta > 0$. Consider the Taylor series of $\kappa_1$ around $z = 0$

$$\kappa_1(z) = \frac{1}{\pi} + \frac{z}{2} + \sum_{n=1}^{\infty} \frac{(2n-3)!!}{(2n-1)n!2^n \pi} z^{2n} \,.$$

We obtain

$$|\kappa_1(z)| \leq \left| \frac{1}{\pi} + \frac{z}{2} \right| + \sum_{n=1}^{\infty} \frac{(2n-3)!!}{(2n-1)n!2^n \pi} |z|^{2n} \leq \frac{1}{2} + \frac{1}{\pi} - \delta_\theta + \sum_{n=1}^{\infty} \frac{(2n-3)!!}{(2n-1)n!2^n \pi} = 1 - \delta_\theta \,.$$

Lemma 17 shows that there exists $0 < \delta' < 1$ such that $\kappa_1^{(k)}$ is analytic on $B_1(\delta') \cap D$. From the argument above, we know that $\kappa_1$ maps $A \triangleq \{z \in \mathbb{C} \mid |z| = 1, |\arg z| \geq \theta\}$ to inside of the open unit ball $B_0(1)$. Since $A$ is compact and Lemma 18 implies that $g$ maps $B_0(1)$ to $B_0(1)$, there exists $1 < R_\theta < 1 + \delta'$ such that $\kappa_1$ maps

$$A_\theta \triangleq (\{z \in \mathbb{C} \mid |z| \leq R_\theta, |\arg z| \geq \theta\} \cap D) \cup B_0(1)$$

to $B_0(1)$. It follows that $\kappa_1^{(k)}$ is analytic on $A_\theta$. Let us pick $\theta \in (0, \pi/2)$ such that $e^{\mathbf{i}\theta} \in B_1(\delta')$. Then we conclude that $\kappa_1^{(k)}$ is analytic on $\{z \in \mathbb{C} \mid |z| \leq R_\theta\} \cap D$. $\qquad\square$

Now we are ready to prove Theorem 7.

*Proof.* Since $\kappa_0$ and $\kappa_1$ are both analytic on $D = \mathbb{C} \setminus [1, \infty) \setminus (-\infty, -1]$, similar arguments as in the proof of Lemma 19 shows that $\kappa_0(\kappa_1^{(k)}(z))$ is analytic on $\{z \in \mathbb{C} \mid |z| \leq R\} \cap D$ for all $k \geq 1$ and some $R > 1$. We then show, for any $k \geq 1$, there exists some $R_k > 1$ such that $N_k(z)$ is analytic on $\{z \in \mathbb{C} \mid |z| \leq R_k\} \cap D$ by induction. The function $N_0(z) = z + \beta^2$ is analytic on $D$. Assume $N_{k-1}(z)$ is analytic on $\{z \in \mathbb{C} \mid |z| \leq R_{k-1}\} \cap D$ for some $R_{k-1} > 1$. Recall that

$$N_k(z) = \kappa_1^{(k)}(z) + N_{k-1}(z)\kappa_0(\kappa_1^{(k-1)}(z)) + \beta^2 \,.$$

Then we can find some $R_k > 1$ such that $N_k(z)$ is analytic on $\{z \in \mathbb{C} \mid |z| \leq R_k\} \cap D$. $\qquad\square$

### A.4 PROOF OF THEOREM 8

*Proof.* We first analyze the behavior of $N_k(z)$ as $z \to 1$ for any $k \geq 1$. We aim to show, for any $k \geq 1$, there exists a sequence of complex functions $p_k(z)$ with $\lim_{z \to 1} p_k(z) = -\sqrt{2}(1+\beta^2)k(k+1)/2\pi$ such that

$$N_k(z) = (k+1)(z+\beta^2) + p_k(z)\sqrt{1-z} \,. \tag{15}$$

We prove by induction on $k$. Recall

$$\kappa_0(z) = \frac{\pi + \mathbf{i} \log(z + \mathbf{i}\sqrt{1-z^2})}{\pi} \,.$$

The fundamental theorem of calculus then gives for any $z \in D$

$$\kappa_0(z) = 1 + \int_\gamma \frac{1}{\pi\sqrt{1-w^2}} dw \,,$$

where $\gamma : [0,1] \to \mathbb{C}$ is the simple straight line connecting $1$ and $z$. As $z \to 1$, we have $\frac{1}{\sqrt{1-z^2}} \sim \frac{1}{\sqrt{2}\sqrt{1-z}}$. Therefore, similar arguments as in the proof of Lemma 14 give

$$\kappa_0(z) = 1 + h(z)\sqrt{1-z}\,,$$

where $\lim_{z \to 1} h(z) = -\frac{\sqrt{2}}{\pi}$. Combining with Lemma 14 further gives, for any $k \geq 1$

$$\kappa_0(\kappa_1^{(k)}(z)) = 1 + h(\kappa_1^{(k)}(z))\sqrt{1 - z - c_k(z)(1-z)^{3/2}} = 1 + h_k(z)\sqrt{1-z}\,,$$

where $\lim_{z \to 1} h_k(z) = -\frac{\sqrt{2}}{\pi}$. For $k = 1$, we then have

$$N_1(z) = \kappa_1(z) + (z + \beta^2)\kappa_0(z) + \beta^2 = z + d_1(z)(1-z)^{3/2} + (z + \beta^2)(1 + h(z)\sqrt{1-z}) + \beta^2$$
$$= 2(z + \beta^2) + p_1(z)\sqrt{1-z}\,,$$

where $\lim_{z \to 1} d_1(z) = \frac{2\sqrt{2}}{3\pi}$ and $\lim_{z \to 1} p_1(z) = -\sqrt{2}(1 + \beta^2)/\pi$. Assume $N_{k-1}(z) = k(z + \beta^2) + p_{k-1}(z)\sqrt{1-z}$ with $\lim_{z \to 1} p_{k-1}(z) = -\sqrt{2}(1 + \beta^2)k(k-1)/(2\pi)$. We further have

$$N_k(z) = \kappa_1^{(k)}(z) + N_{k-1}(z)\kappa_0(\kappa_1^{(k-1)}(z)) + \beta^2$$
$$= z + d_k(z)(1-z)^{3/2} + \left(k(z + \beta^2) + p_{k-1}(z)\sqrt{1-z}\right)(1 + h_{k-1}(z)\sqrt{1-z}) + \beta^2$$
$$= (k+1)(z + \beta^2) + (p_{k-1}(z) + k \cdot h_{k-1}(z)(z + \beta^2))\sqrt{1-z}$$
$$= (k+1)(z + \beta^2) + p_k(z)\sqrt{1-z}\,.$$

where we set $p_k(z) = p_{k-1}(z) + k \cdot h_{k-1}(z)(z + \beta^2)$ and $d_k(z) \to \frac{2\sqrt{2}k}{3\pi}$, $h_{k-1}(z) \to -\frac{\sqrt{2}}{\pi}$ as $z \to 1$. Moreover, we have

$$\lim_{z \to 1} p_k(z) = \lim_{z \to 1} \left\{ p_{k-1}(z) + k \cdot h_{k-1}(z)(z + \beta^2) \right\}$$
$$= -\frac{\sqrt{2}(1 + \beta^2)k(k-1)}{2\pi} - k \cdot \frac{\sqrt{2}}{\pi}(1 + \beta^2)$$
$$= -\frac{\sqrt{2}(1 + \beta^2)k(k+1)}{2\pi}\,,$$

which is desired. This proves (15).

Next we study the behavior of $N_k(z)$ as $z \to -1$ for any $k \geq 1$. We aim to show, for any $k \geq 1$, there exists a sequence of complex functions $q_k(z)$ with $\lim_{z \to -1} q_k(z) = \sqrt{2}(\beta^2 - 1)\prod_{j=1}^{k-1}\kappa_0(a_j)/\pi$ and $a_k \triangleq \kappa_1^{(k)}(-1)$ as defined in Lemma 15 such that

$$N_k(z) = N_k(-1) + q_k(z)\sqrt{1+z}\,. \tag{16}$$

We again adopt induction on $k$. Taylor's theorem gives

$$\kappa_0(z) = \kappa_0(a_k) + r_k(z)(z - a_k)\,,$$

where $\lim_{z \to a_k} r_k(z) = \kappa_0'(a_k) > 0$. Combining with Lemma 15 further gives, for any $k \geq 1$

$$\kappa_0(\kappa_1^{(k)}(z)) = \kappa_0(a_k) + r_k(\kappa_1^{(k)}(z))b_k(z)(z+1)^{3/2} = \kappa_0(a_k) + \tilde{r}_k(z)(z+1)^{3/2}\,,$$

where $b_k(z) \to \frac{2\sqrt{2}}{3\pi}\prod_{j=1}^{k-1}\kappa_1'(a_k)$ and $\tilde{r}_k(z) \to \frac{2\sqrt{2}}{3\pi}\kappa_0'(a_k)\prod_{j=1}^{k-1}\kappa_1'(a_k) > 0$ as $z \to -1$ by Lemma 15. For $k = 1$, the fundamental theorem of calculus gives for any $z \in D$

$$\kappa_0(z) = \int_\gamma \frac{1}{\pi\sqrt{1-w^2}}dw\,,$$

where $\gamma : [0,1] \to \mathbb{C}$ is the simple straight line connecting $-1$ and $z$. As $z \to -1$, we have $\frac{1}{\sqrt{1-z^2}} \sim \frac{1}{\sqrt{2}\sqrt{1+z}}$. Therefore, similar arguments as in the proof of Lemma 14 give

$$\kappa_0(z) = g(z)\sqrt{1+z}\,,$$

where $g(z) \to \frac{\sqrt{2}}{\pi}$ as $z \to -1$. We then have

$$
\begin{aligned}
N_1(z) &= \kappa_1(z) + (z + \beta^2)\kappa_0(z) + \beta^2 \\
&= a_1 + b_1(z)(z+1)^{3/2} + (z + \beta^2)g(z)\sqrt{1+z} + \beta^2 \\
&= (a_1 + \beta^2) + q_1(z)\sqrt{1+z} \\
&= N_1(-1) + q_1(z)\sqrt{1+z} \,,
\end{aligned}
$$

where $N_1(-1) = a_1 + \beta^2 \lim_{z \to -1} q_1(z) = \frac{\sqrt{2}}{\pi}(\beta^2 - 1)$. Assume $N_{k-1}(z) = N_{k-1}(-1) + q_{k-1}(z)\sqrt{1+z}$ with $\lim_{z \to -1} q_{k-1}(z) = \sqrt{2}(\beta^2 - 1)\prod_{j=1}^{k-2}\kappa_0(a_j)/\pi$. We further have

$$
\begin{aligned}
N_k(z) &= \kappa_1^{(k)}(z) + N_{k-1}(z)\kappa_0(\kappa_1^{(k-1)}(z)) + \beta^2 \\
&= a_k + b_k(z)(z+1)^{3/2} + N_{k-1}(z)\left(\kappa_0(a_{k-1}) + \tilde{r}_{k-1}(z)(z+1)^{3/2}\right) + \beta^2 \\
&= \left(a_k + \beta^2 + N_{k-1}(z)\kappa_0(a_{k-1})\right) + \left(b_k(z) + N_{k-1}(z)\tilde{r}_{k-1}(z)\right)(z+1)^{3/2} \\
&= \left(a_k + \beta^2 + N_{k-1}(-1)\kappa_0(a_{k-1})\right) + q_{k-1}(z)\kappa_0(a_{k-1})\sqrt{z+1} \\
&\quad + \left(b_k(z) + N_{k-1}(z)\tilde{r}_{k-1}(z)\right)(z+1)^{3/2} \\
&= N_k(-1) + q_{k-1}(z)\kappa_0(a_{k-1})\sqrt{z+1} + \left(b_k(z) + N_{k-1}(z)\tilde{r}_{k-1}(z)\right)(z+1)^{3/2} \\
&= N_k(-1) + q_k(z)\sqrt{1+z} \,,
\end{aligned}
$$

where we use the induction assumption in the fourth equation, use the fact $N_k(-1) = a_k + \beta^2 + N_{k-1}(-1)\kappa_0(a_{k-1})$ in the fifth equation and define

$$
q_k(z) = q_{k-1}(z)\kappa_0(a_{k-1}) + \left(b_k(z) + N_{k-1}(z)\tilde{r}_{k-1}(z)\right)(z+1)
$$

in the last equation. We also have

$$
\begin{aligned}
\lim_{z \to -1} q_k(z) &= \lim_{z \to -1}\left\{q_{k-1}(z)\kappa_0(a_{k-1}) + \left(b_k(z) + N_{k-1}(z)\tilde{r}_{k-1}(z)\right)(z+1)\right\} \\
&= \lim_{z \to -1}\left\{q_{k-1}(z)\kappa_0(a_{k-1})\right\} \\
&= \frac{\sqrt{2}(\beta^2 - 1)}{\pi}\prod_{j=1}^{k-1}\kappa_0(a_j)\,,
\end{aligned}
$$

which is desired. This proves (16).

Finally, according to Theorem 7, combining (15) and (16), applying (Flajolet & Sedgewick, 2009, Theorem VI.5) with $\rho = 1, r = 2, \tau(z) = (1-z)^{1/2}, \zeta_1 = 1, \zeta_2 = -1, \sigma_1(z) = (k+1)(z+\beta^2)$, $\sigma_2(z) = N_k(-1), \mathbf{D} = \{z \in \mathbb{C} \mid |z| \le R_k\} \cap D$, we conclude $[z^n]N_k(z) = O(n^{-3/2})$. □

## B  PROOFS FOR EXPONENTIAL POWER KERNEL

### B.1  PROOF OF LEMMA 9

*Proof.* According to (Doetsch, 1974, Theorem 28.2), we have, for $0 < a < 1$,

$$
f(t) = \frac{1}{2\pi \mathbf{i}} \lim_{T \to +\infty} \int_{x_0 - \mathbf{i}T}^{x_0 + \mathbf{i}T} \exp(ts - s^a)ds \qquad (x_0 \ge 0)\,.
$$

Also (Doetsch, 1974, Theorem 28.2) implies that $f(t)$ is continuous in $-\infty < t < +\infty$ and $f(0) = 0$.

Next we explicitly calculate $f(t)$ using Bromwich contour integral. We denote each part of the Bromwich contour by $\Gamma_0, \ldots, \Gamma_5$ as depicted in Fig. 2. Denote the radius of the outer and inner arc by $R$ and $r$. When $T \to \infty$, we have $R = \sqrt{T^2 + x_0^2} \to \infty$. Also we let $r \to 0$ and $\Gamma_2, \Gamma_4$ tend to $(-\infty, 0]$ from above and below respectively in the limit. By the residue theorem, we have

$$
\left(\int_{\Gamma_0} + \ldots + \int_{\Gamma_5}\right)\exp(ts - s^a)ds = 0\,,
$$

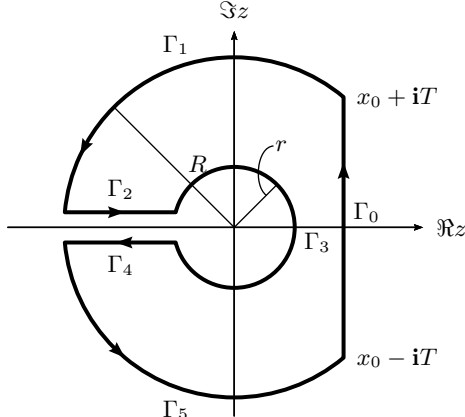

Figure 2: Bromwich contour that circumvents the branch cut $(-\infty, 0]$

which implies

$$
\begin{aligned}
\lim_{T \to \infty} \int_{x_0 - \mathbf{i}T}^{x_0 + \mathbf{i}T} \exp(ts - s^a)ds &= \lim_{T \to \infty} \int_{\Gamma_0} \exp(ts - s^a)ds \\
&= -\lim \left( \int_{\Gamma_1} + \ldots + \int_{\Gamma_5} \right) \exp(ts - s^a)ds \\
&\triangleq -\lim(I_1 + \ldots + I_5),
\end{aligned}
$$

where the last two limits are taken as $R \to \infty$, $r \to 0$, and $\Gamma_2, \Gamma_4$ tend to $(-\infty, 0]$. We then calculate each part separately.

**Part I:** We calculate the parts for $\Gamma_1$ and $\Gamma_5$. We follow the similar idea as in the proof of (Spiegel, 1965, Theorem 7.1). Along $\Gamma_1$, since $s = Re^{\mathbf{i}\theta}$ with $\theta_0 \leq \theta \leq \pi$, $\theta_0 = \arccos(x_0/R)$,

$$
\begin{aligned}
I_1 &= \int_{\theta_0}^{\pi/2} e^{Re^{\mathbf{i}\theta}t} e^{-R^a e^{\mathbf{i}a\theta}} \mathbf{i}Re^{\mathbf{i}\theta}d\theta + \int_{\pi/2}^{\pi} e^{Re^{\mathbf{i}\theta}t} e^{-R^a e^{\mathbf{i}a\theta}} \mathbf{i}Re^{\mathbf{i}\theta}d\theta \\
&\triangleq I_{11} + I_{12}.
\end{aligned}
$$

For $I_{11}$,

$$
\begin{aligned}
|I_{11}| &\leq \int_{\theta_0}^{\pi/2} |e^{Rt\cos\theta}| \cdot |e^{-R^a \cos(a\theta)}| Rd\theta \\
&\leq \int_{\theta_0}^{\pi/2} e^{Rt\cos\theta} \cdot e^{-R^a \cos(a\pi/2)} Rd\theta \\
&\leq \frac{R}{R^a \cos(a\pi/2)} \int_{\theta_0}^{\pi/2} e^{Rt\cos\theta} d\theta \\
&= \frac{R}{R^a \cos(a\pi/2)} \int_0^{\phi_0} e^{Rt\sin\phi} d\phi,
\end{aligned}
$$

where $\phi_0 = \pi/2 - \theta_0 = \arcsin(x_0/R)$. Since $\sin\phi \leq \sin\phi_0 \leq x_0/R$, we have

$$
|I_{11}| \leq \frac{R}{R^a \cos(a\pi/2)} \phi_0 e^{x_0 t} = \frac{R}{R^a \cos(a\pi/2)} e^{x_0 t} \arcsin(x_0/R).
$$

As $R \to \infty$, we have $\lim_{R \to \infty} I_{11} = 0$.

For $I_{12}$,

$$
|I_{12}| \leq \int_{\pi/2}^{\pi} e^{Rt\cos\theta} \cdot e^{-R^a \cos(a\theta)} Rd\theta.
$$

First, we consider the case $0 < a < 1/2$. We have $a\theta \le a\pi < \pi/2$ and $\cos(a\theta) \ge \cos(a\pi) > 0$. It follows

$$\int_{\pi/2}^{\pi} e^{Rt\cos\theta} \cdot e^{-R^a\cos(a\theta)} Rd\theta$$

$$\le Re^{-R^a\cos(a\pi)} \int_{\pi/2}^{\pi} e^{Rt\cos\theta} d\theta$$

$$= Re^{-R^a\cos(a\pi)} \int_{0}^{\pi/2} e^{-Rt\sin\phi} d\phi$$

$$\le Re^{-R^a\cos(a\pi)} \int_{0}^{\pi/2} e^{-2Rt\phi/\pi} d\phi$$

$$= e^{-R^a\cos(a\pi)} \frac{\pi(1-e^{-Rt})}{2t},$$

where in the last inequality we use the fact $\sin\phi \ge 2\phi/\pi$ for $\phi \in [0, \pi/2]$. Thus, $\lim_{R\to\infty} I_{12} = 0$. Next, we consider $1/2 \le a < 1$. Define

$$p(\theta) \triangleq Rt\cos\theta - R^a\cos(a\theta).$$

We then have its second derivative as follows

$$p''(\theta) = a^2 R^a \cos(a\theta) - Rt\cos(\theta).$$

Choose $\delta$ to be a fixed constant in $(0, \frac{\pi}{2}(\frac{1}{a}-1))$. Since $a \ge 1/2$, then $\delta < \pi/2$. If $\pi/2+\delta \le \theta \le \pi$,

$$p''(\theta) \ge -a^2 R^a - Rt\cos(\pi/2+\delta) = -a^2 R^a + Rt\sin(\delta).$$

Since $a < 1$, there exists some large $R_1 > 0$ such that $p''(\theta) \ge -a^2 R^a + Rt\sin(\delta) > 0$ holds for all $R > R_1$. If $\pi/2 \le \theta < \pi/2 + \delta$,

$$p''(\theta) \ge a^2 R^a \cos(a(\pi/2+\delta)).$$

Since $a(\pi/2 + \delta) < \pi/2$ by the choice of $\delta$, we get $\cos(a(\pi/2 + \delta)) > 0$. Then we also have $p''(\theta) > 0$. Therefore, if $R > R_1$, $p(\theta)$ is convex in $\theta \in [\pi/2, \pi]$. As a result, we get

$$\max_{\theta \in [\pi/2, \pi]} p(\theta) \le \max\{p(\pi/2), p(\pi)\}.$$

Write

$$h(R, \theta) \triangleq Re^{Rt\cos\theta} \cdot e^{-R^a\cos(a\theta)} = Re^{p(\theta)}.$$

Then we have

$$\max_{\theta \in [\pi/2, \pi]} h(R, \theta) \le \max\{h(R, \pi/2), h(R, \pi)\}$$

$$= R\max\{e^{-R^a\cos(\frac{\pi a}{2})}, e^{-R^a\cos(\pi a)-Rt}\}$$

$$\le R\max\{e^{-R^a\cos(\frac{\pi a}{2})}, e^{R^a-Rt}\},$$

which goes to 0 as $R \to \infty$. Therefore, $h(R, \theta)$ converges to 0 uniformly (as a function of $\theta \in [\pi/2, \pi]$ with index $R$), which implies

$$\lim_{R\to\infty} \int_{\pi/2}^{\pi} h(R, \theta)d\theta = 0.$$

Hence, we establish $\lim_{R\to\infty} I_{12} = 0$ for all $a \in (0, 1)$.

Combining these above, we conclude $\lim_{R\to\infty} I_1 = 0$. Similarly, $\lim_{R\to\infty} I_5 = 0$.

**Part II:** We calculate the parts for $\Gamma_2$ and $\Gamma_4$. By the dominated convergence theorem, we have, for $y > 0$

$$\lim_{\substack{R\to\infty \\ r\to 0 \\ y\to 0^+}} I_2 = \lim_{\substack{R\to\infty \\ r\to 0 \\ y\to 0^+}} \int_{-R+iy}^{-r+iy} \exp(ts)\exp(-s^a)ds$$

$$= \lim_{\substack{R\to\infty \\ r\to 0 \\ y\to 0^+}} \int_{-R+iy}^{-r+iy} \exp(ts) \sum_{k=0}^{\infty} \frac{(-1)^k s^{ak}}{k!}ds$$

$$= \lim_{\substack{R\to\infty \\ r\to 0 \\ y\to 0^+}} \sum_{k=0}^{\infty} \frac{(-1)^k}{k!} \int_{-R+iy}^{-r+iy} \exp(ts)s^{ak}ds\,.$$

We then calculate the limit of the summand.

$$\lim_{\substack{R\to\infty \\ r\to 0 \\ y\to 0^+}} \int_{-R+iy}^{-r+iy} \exp(ts)s^{ak}ds = \int_{-\infty}^{0} e^{tx} \cdot [(-x)e^{i\pi}]^{ak}dx$$

$$= \int_{0}^{\infty} e^{-tx} x^{ak} e^{i\pi ak}dx$$

$$= \frac{1}{t^{ak+1}}\Gamma(ak+1)e^{i\pi ak}\,.$$

Similarly, we obtain the corresponding part in $\Gamma_4$:

$$\lim_{\substack{R\to\infty \\ r\to 0 \\ y\to 0^-}} \int_{-r+iy}^{-R+iy} \exp(ts)s^{ak}ds = -\int_{-\infty}^{0} e^{tx} \cdot [(-x)e^{-i\pi}]^{ak}dx$$

$$= -\frac{1}{t^{ak+1}}\Gamma(ak+1)e^{-i\pi ak}\,.$$

Combining the parts of $\Gamma_2$ and $\Gamma_4$ together, we get

$$\lim(I_2 + I_4) = \sum_{k=0}^{\infty} \frac{(-1)^k}{k!} \frac{2i\Gamma(ak+1)\sin(\pi ak)}{t^{ak+1}}\,.$$

**Part III:** We get the limit for $\Gamma_3$ is 0 as $r \to 0$.

Combining the three parts above, we conclude

$$f(t) = \frac{1}{2\pi i} \sum_{k=0}^{\infty} \frac{(-1)^{k+1}}{k!} \frac{2i\Gamma(ak+1)\sin(\pi ak)}{t^{ak+1}}$$

$$= \frac{1}{\pi} \sum_{k=0}^{\infty} \frac{(-1)^{k+1}\Gamma(ak+1)\sin(\pi ak)}{k!t^{ak+1}}\,.$$

$\square$

## B.2 PROOF OF LEMMA 10

*Proof.* Euler's reflection formula gives

$$\Gamma(1+ka)\Gamma(-ka) = \frac{-\pi}{\sin(\pi ka)}, \quad ka \notin \mathbb{Z}\,.$$

According to Lemma 9, we have

$$\begin{aligned}
f(t) &= \frac{1}{\pi} \sum_{k=0}^{\infty} \frac{(-1)^{k+1} \Gamma(ak+1) \sin(\pi a k)}{k! t^{ak+1}} \\
&= \sum_{k=0}^{\infty} \frac{(-1)^k}{k! t^{ak+1} \Gamma(-ak)} \\
&= \sum_{j=1}^{q-1} \sum_{n=0}^{\infty} \frac{(-1)^{nq+j}}{(nq+j)! t^{a(nq+j)+1} \Gamma(-a(nq+j))} \, .
\end{aligned} \tag{17}$$

First, we show that the series in (17) converges absolutely:

$$\begin{aligned}
&\sum_{j=1}^{q-1} \sum_{n=0}^{\infty} \frac{|t|^{-a(nq+j)-1}}{(nq+j)! |\Gamma(-a(nq+j))|} \\
&= \sum_{j=1}^{q-1} \frac{1}{|t|^{aj+1}} \sum_{n=0}^{\infty} \frac{|t|^{-np}}{(nq+j)! |\Gamma(-a(nq+j))|} \\
&= \sum_{j=1}^{q-1} \frac{1}{|t|^{aj+1} |\Gamma(-aj)|} \sum_{n=0}^{\infty} \frac{|t|^{-np} \prod_{i=1}^{np} (aj+i)}{(nq+j)!} \, .
\end{aligned} \tag{18}$$

The inner summation in (18) is a power series in $|t|^{-p}$. We would like to show that its radius of convergence is $\infty$. Define

$$b_n = \frac{\prod_{i=1}^{np}(aj+i)}{(nq+j)!} \, .$$

We have

$$\begin{aligned}
\frac{b_{n+1}}{b_n} &= \frac{\prod_{np < i \le (n+1)p}(aj+i)}{\prod_{nq < i \le (n+1)q}(j+i)} = \frac{\prod_{i=1}^{p} \frac{aj+np+i}{j+nq+i}}{\prod_{i=nq+p+1}^{(n+1)q}(j+i)} \\
&\le \frac{1}{\prod_{i=nq+p+1}^{(n+1)q}(j+i)} \le \frac{1}{(j+nq+p+1)^{q-p}} \to 0 \, .
\end{aligned}$$

As a result, the radius of convergence is $\infty$. Then we have

$$\begin{aligned}
f(t) &= \sum_{j=1}^{q-1} \frac{1}{t^{aj+1} \Gamma(-aj)} \sum_{n=0}^{\infty} \frac{(-1)^{n(p+q)+j} t^{-pn} \prod_{i=1}^{np}(aj+i)}{(nq+j)!} \\
&= \sum_{j=1}^{q-1} \frac{1}{t^{aj+1} \Gamma(-aj)} \left( \frac{(-1)^j}{j!} + \underbrace{\sum_{n=1}^{\infty} \frac{(-1)^{n(p+q)+j} t^{-pn} \prod_{i=1}^{np}(aj+i)}{(nq+j)!}}_{A} \right)
\end{aligned}$$

Notice that the quantity $A$ goes to 0 as $t \to +\infty$. Therefore we deduce

$$f(t) \sim \sum_{j=1}^{q-1} \frac{(-1)^j}{t^{aj+1} j! \Gamma(-aj)} \sim -\frac{1}{t^{a+1} \Gamma(-a)} \, ,$$

as $t \to +\infty$. □

