# OpenReview forum: "Deep Neural Tangent Kernel and Laplace Kernel Have the Same RKHS"
_ICLR.cc/2021/Conference — ICLR 2021 Poster_

### Official Review · AnonReviewer3 · 2020-10-26
**strong article: RKHS of NTK in deep ReLU nets = Laplace kernel on the unit sphere**

**Rating:** 8
**Confidence:** 4

**Review:**

This paper uses singularity analysis developed in the context of analytic combinatorics to study the relationship between the reproducing kernel Hilbert spaces of the NTK in a deep fully connected ReLU network, the Laplace kernel, and exponential power kernels. The main results are when these kernels are restricted to the unit sphere. In particular, the authors show that, as vector spaces, the RKHS on the unit sphere of the NTK for a ReLU network of any fixed depth are the same and in fact coincides with that of the Laplace kernel. The authors also compare the RKHS for exponential power kernels, demonstrating that larger powers lead to smaller Hilbert spaces.

Strong Points:
1. Understanding the NTK of a deep ReLU networks is a popular and important topic.
2. The NTK is defined via non-linear recursions, meaning that obtaining quantitative results for deep networks is both technically and conceptually challenging.
3. Proving that the RKHS of the Laplace and ReLU kernels coincides as vectors spaces is a substantial result.
4. The authors are the first, to my knowledge, to apply the rather elegant singularity analysis of Flajolet-Sedgewick to study the NTK. Perhaps the closes prior work used free probability to study the spectrum of the NTK.
5. The paper is clearly written.

Weak Points:
1. As the authors themselves allude to in the discussion, showing that two RKHS are the same as vector spaces ignores the Hilbert spaces structure and can still lead to different inductive biases in kernel regression.
2. The present work applies only to ReLU networks, while the recursive definition of the NTK is valid for rather general non-linearities.

Overall, this article is a solid theoretical contribution to our understanding of the NTK. In addition to proving concrete results comparing the Laplace, NTK, and exponential power kernels, it is serves as a proof-of-concept for potentially using the tools of singularity analysis to understand neural networks.

---

> ### Author Response · Authors · 2020-11-17
> **Response to Reviewer 3**
>
> Thank you so much for your time, insightful feedback, positive evaluation :-) We have incorporated the changes into the revised paper, and address the issues in detail here:
>
> Q: As the authors themselves allude to in the discussion, showing that two RKHS are the same as vector spaces ignores the Hilbert spaces structure and can still lead to different inductive biases in kernel regression.
>
> A: Thank you for the great point! The follow-up work that we are working on shows that we can also derive the RKHS norm from the singularity analysis and the RKHS norm reflects the structure of the Hilbert space.
>
> Q: The present work applies only to ReLU networks, while the recursive definition of the NTK is valid for rather general non-linearities.
>
> A: This is a very interesting direction for future work, which we are also considering. We would like to remark that the ReLU NTK has a clean expression partly because the expectation over the Gaussian process in the general NTK can be computed exactly if the activation function is ReLU (which may not be true for other non-linearities, for example, it may require more work for sigmoid). We further discussed it in the revised Section 6.

---

### Official Review · AnonReviewer2 · 2020-10-28
**Review on the theoretical analysis of the relationship between Laplace kernel and NTK**

**Rating:** 7
**Confidence:** 4

**Review:**

This paper proves that the reproducing kernel Hilbert spaces of a deep neural tangent kernel and the Laplace kernel have the same set of functions when they restricted to the sphere $S^{d-1}$, which improves the results established in Geifman et al., 2020. Moreover, the paper proves that more non-smooth of the exponential power kernel leads to a larger RKHS with restriction on the sphere $S^{d-1}$ and the entire $R^d$. Furthermore, the authors conduct numerical experiments to verify the asymptotics of the Maclaurin coefficients of the Laplace kernel and NTKs kernel. In summary, the paper is well-written and organized logically. The proof of theoretical results of this paper seems to be correct and reasonable, resulting from the full details of the proof provided in the appendices.
The contribution of this paper includes two parts. Firstly, it aims to explain why the Laplace kernel and NTK have similar performance in experiments from theoretical point of view by showing that the space of the Laplace kernel and NTK are the same when limited to a sphere. On the other hand, the author reveals the relationship between the smoothness of the exponential power kernel and the corresponding RKHS to explain the better performance of the exponential kernel with a smaller power in the experiments.
Now I would like to give some comments, 1. Do Laplace kernel and NTK have a similar learning dynamic when we perform kernelized gradient decent in real-world dataset? 2. It is necessary to study the behavior of the NTK and the Laplace kernel outside of $S^{d-1}$. I wonder whether the theoretical results proposed are helpful to improve the performance of Laplace kernel or NTK. 3. The author demonstrates that a non-smooth exponential power kernel leads to a larger RKHS, but whether this indicates that the model obtained by adopting a non-smooth kernel has greater generalization capability. I think the author needs to further theoretically illustrate the relationship between them.

---

> ### Author Response · Authors · 2020-11-17
> **Response to Reviewer 2**
>
> Thank you so much for your time, insightful feedback, positive evaluation :-) We have incorporated the changes into the revised paper, and address the issues in detail here:
>
> Q: Do Laplace kernel and NTK have a similar learning dynamic when we perform kernelized gradient decent in real-world dataset?
>
> A: Thank you for asking this meaningful question. Since the main focus of our paper lies in the theoretical side, we have not performed these two methods on the real datasets. We leave this to future work. To the best of our knowledge, Belkin et al. [1] showed the Laplace kernel has similar performance in fitting random labels as neural networks. Geifman et al. [2] also found the similar performance of the Laplace kernel and the NTK on some real datasets such as UCI[3], MillionSongs[4], SUSY[5], and HIGGS[5].
>
> [1] Mikhail Belkin, Siyuan Ma, and Soumik Mandal. To understand deep learning we need to understand kernel learning. In International Conference on Machine Learning, pp. 541–549, 2018.
>
> [2] Amnon Geifman, Abhay Yadav, Yoni Kasten, Meirav Galun, David Jacobs, and Ronen Basri. On the similarity between the laplace and neural tangent kernels. arXiv preprint arXiv:2007.01580, 2020.
>
> [3] Manuel Fern´andez-Delgado, Eva Cernadas, Sen´en Barro, and Dinani Amorim. Do we need hundreds of classifiers to solve real world classification problems? The Journal of Machine Learning Research, 15(1): 3133–3181, 2014.
>
> [4] Thierry Bertin-Mahieux, Daniel P.W. Ellis, Brian Whitman, and Paul Lamere. The million song dataset. In Proceedings of the 12th International Conference on Music Information Retrieval (ISMIR), 2011.
>
> [5] Pierre Baldi Peter Sadowski and Daniel Whiteson. Searching for exotic particles in high-energy physics with deep learning. Nature communications, 5, 2014.
>
> Q: It is necessary to study the behavior of the NTK and the Laplace kernel outside of $S^{d-1}$. I wonder whether the theoretical results proposed are helpful to improve the performance of Laplace kernel or NTK.
>
> A: Thank you for the suggestion! We made some partial results on the behavior of the NTK and the Laplace kernel outside of the unit sphere, which shows that the ReLU NTK is equivalent to a homogeneous Laplace kernel in the whole space. We will present these results in our follow-up work.
>
> Q: The author demonstrates that a non-smooth exponential power kernel leads to a larger RKHS, but whether this indicates that the model obtained by adopting a non-smooth kernel has greater generalization capability. I think the author needs to further theoretically illustrate the relationship between them.
>
> A: Thank you for pointing out this interesting question. The relationship between them is as follows. Our results show that using a more non-smooth exponential power kernel leads to a larger RKHS and therefore greater expressive power. However, analyzing the generalization error requires more efforts in general. Researchers often use the RKHS norm to provide an upper bound for it. The follow-up work that we are working on shows that we can also derive the RKHS norm from the singularity analysis and thereby give an upper bound on the generalization error. We clarified this in our revised Section 6.

---

### Official Review · AnonReviewer4 · 2020-10-28

**Rating:** 7
**Confidence:** 3

**Review:**

This paper compares the reproducing kernel Hilbert spaces (RKHSes) generated by a deep neural tangent kernel (NTK), a Laplacian kernel, and an exponential power kernel. The main result is twofold: 1) the RKHSes associated with a NTK and a Laplacian kernel  contain the same set of functions when restricted to the sphere S^{d-1}; and 2) the RKHS generated by the exponential kernel with a smaller power contains more functions when restricted to the sphere S^{d-1} or R^d.

The key proof idea involves an analysis  the asymptotic rates of the Maclaurin coefficients of the kernels. Aronszajn’s lemma is then used to establish the inclusion of the RKHS. A notable contribution in the proof is the application of analytic combinatorics to obtain the asymptotic rate O(n^{-3/2}) of the Maclaurin coefficients of the NTK which is intractable due to its recursive definition.  The authors also provide numerical results showing the empirical rate matches the theoretical asymptotic rate of the Maclaurin coefficients of the Laplace kernel and NTKs.

Overall, I think the paper is well-written, and the proof is solid.  The proof technique using analytic combinatorics seems novel, and the main results are useful for theoretical understanding of neural networks.

---

> ### Author Response · Authors · 2020-11-17
> **Response to Reviewer 4**
>
> Thank you so much for your time, insightful feedback, and positive evaluation :-)

---

### Official Review · AnonReviewer1 · 2020-10-29
**A clearly written paper but I still have some doubts**

**Rating:** 5
**Confidence:** 3

**Review:**

The goal of this paper is to complete the theoretical subset inclusion relationships given in (Geifman et al., 2020).
The authors proved that the RKHS of the the neural tangent kernel (NTK) with any number of layers
is a subset of the RKHS of the Laplace kernel.
Combined with the results in (Geifman et al., 2020),
if the input domain is restricted to the sphere of $(d-1)$-dimensional real space,
the RKHS of NTK with any number of layers
is equal to the RKHS of the Laplace kernel.

The writing of this paper is very clear.
I can easily get what the authors intend to express.
However, I still have some doubts about the details.

In Section 2.1,
the original Laplace kernels and exponential power kernels are all classic shift-invariant kernels.
Here the authors provided the dot-product versions of these two kernels.
Why do the authors need to do these changes?
For the sphere $\mathbb{S}^{d-1}$, the shift-invariant versions are equivalent to the dot-product kernels?

Under Lemma 3,
if $\Sigma_k(x, x) = 1$,
we can only get $\Sigma_k(u) = \Sigma_{k-1}(u)$.
What does the notation $\kappa_1^{(k)}$ means here?

Theorem 1 in (Aronszajn, 1950, p. 354) just stated that if $K_1 \preccurlyeq K_2$, then $\mathcal{K}_1 \subseteq \mathcal{K}_2$.
I am very curious and doubting about Lemma 4.
For any positive constant $\gamma > 0$,
does Lemma 4 always exist?

Under Lemma 4, the authors stated that:
1. "Then the Maclaurin series of $K_{\mathrm{Lap}}(u)$ and $N_k(u)$ have all non-negative coefficients by the classical approximation theory."
2. "If the Maclaurin series of $K(u)$ have all non-negative coefficients,
$K$ is a positive definite kernel on the unit sphere.

I cannot find the original documents of (Schoenberg, 1942) and (Cheney and Light, 2009).
Could you please provide these results in the Appendix such that we can check the correctness of these results?
This is because, in the following part, all the proofs will be focused on $\gamma^{2}\left[z^{n}\right] K_{\text {Lap }}(z) \geq\left[z^{n}\right] N_{k}(z)$.

In Section 3,
I have two doubting points.
1. The first one is under the definition of $\Delta$-domain.
The authors used $z$ to replace $u$ and $u=x^{\top} y$ as shown Section 2.1.
If we put $u=x^{\top} y = 1$,
the Laplace kernels and exponential power kernels given in Section 2.1 will all become constant.
Could the authors explain more about this assumption?
2. The second one is under Lemma 5.
The authors stated that "Careful singularity analysis then gives ...".
It is very difficult for me to easily get these two results about Maclaurin coefficients.
Could you give the detailed steps for the derivation of these results?

In Theorem 2,
I am very curious about the seemingly contrary results:
$\mathcal{H}_{K_{\text {exp }}^{\gamma_1, \sigma_1}} \left(\mathbb{S}^{d-1}\right) \subseteq \mathcal{H}_{K_{\text {exp }}^{\gamma_2, \sigma_2}}\left(\mathbb{S}^{d-1}\right)$, and
$\mathcal{H}_{K_{\mathrm{exp}}^{\gamma_2, \sigma_2}}\left(\mathbb{R}^{d}\right) \subseteq \mathcal{H}_{K_{\mathrm{exp}}^{\gamma_1, \sigma_1}}\left(\mathbb{R}^{d}\right).$
Could you provide more explanations about this point in Section 4.
The superficial explanations cannot help us understand this point.
We need to go deeper into the proof sketch to figure out what makes this phenomenon happen.

I didn't go much deeper into the Appendix,
so it is hard for me to check the correctness of the whole paper.
Since ICLR is a highly selective conference,
I will give a score 5 at present.
After the authors solved my concerns and other reviewers verifies the correctness of all proofs,
I may raise my score to 6 or 7.

---

> ### Author Response · Authors · 2020-11-17
> **Response to Reviewer 1**
>
> Thank you so much for your time and insightful feedback! We have incorporated the changes into the revised paper, and address the issues in detail here:
>
> Q: In Section 2.1, the original Laplace kernels and exponential power kernels are all classic shift-invariant kernels. Here the authors provided the dot-product versions of these two kernels. Why do the authors need to do these changes? For the sphere, the shift-invariant versions are equivalent to the dot-product kernels?
>
> A: Yes, on the sphere, the shift-invariant kernels are dot product kernels. If $k(\|x-y\|)$ is a shift-invariant kernel, then we have $k(\|x-y\|)=k(\sqrt{\|x-y\|^2}) = k(\sqrt{\|x\|^2+\|y\|^2-2x^\top y}) = k(\sqrt{2-2x^\top y})$. We use $\|x\|=\|y\|=1$ ($x$ and $y$ are on the sphere) in the last equality. We can see that it is a function of $x^\top y$ and therefore a dot-product kernel. We expressed it as a dot-product kernel in order to analyze the decay rate of its power series coefficients and determine its RKHS.
>
> Q: Under Lemma 3, if $\Sigma_k(x,x)=1$, we can only get $\Sigma_k(u)=\Sigma_{k-1}(u)$.
>
> A: Thank you for the great point. Plugging $\Sigma_k(x,x)=1$ into Eq. (2), we get $\Sigma_k(u) = \kappa_1(\Sigma_{k-1}(u))$. By induction, we obtain $\Sigma_k(u) = \kappa_1^{(k)}(u)$. We present a detailed derivation of Eq.(4) in Appendix A.2. in the revision.
>
> Q: What does the notation $\kappa_1^{(k)}$ means here?
>
> A: Thank you for pointing it out. We define it right below Eq. (4). It means the $k$-th iterated composition of $\kappa_1(u)$. For example, $\kappa_1^{(0)}(u)=u$, $\kappa_1^{(1)}(u)=\kappa_1(u)$ and $\kappa_1^{(2)}(u) = \kappa_1(\kappa_1(u))$. We clarified it in the revision.
>
> Q: Theorem 1 in (Aronszajn, 1950, p. 354) just stated that if $K_1\preccurlyeq K_2$, then $\mathcal{K}_1\subseteq \mathcal{K_2}$. I am very curious and doubting about Lemma 4. For any positive constant $\gamma>0$, does Lemma 4 always exist?
>
> A: Theorem 2.17 of Saitoh & Sawano (the second reference in the parentheses) states exactly Lemma 4. In fact, Aronszajn’s version is equivalent because $\gamma^2 \mathcal{K}_2$ has the same RKHS as $\mathcal{K}_2$ (the positive constant does not influence the RKHS). We cited (Aronszajn 1950) because to the best of our knowledge, it is the original literature.
>
> Q: I cannot find the original documents of (Schoenberg, 1942) and (Cheney and Light, 2009). Could you please provide these results in the Appendix such that we can check the correctness of these results?
>
> A: Thank you for the great point. In Lemma 5 in the revision, we restated these results from the original documents.
>
> Q: In Section 3, I have two doubting points. The first one is under the definition of $\Delta$-domain. The authors used $z$ to replace $u$ and $u=x^{\top}y$ as shown Section 2.1. If we put $u=x^{\top}y=1$, the Laplace kernels and exponential power kernels given in Section 2.1 will all become constant. Could the authors explain more about this assumption?
>
> A: We are sorry for this confusion. The condition that $x$ and $y$ live on the unit sphere means $x^{\top}x=y^{\top}y=1$. This cannot imply $x^{\top}y=1$. Actually $x^{\top}y$ can be any real number in the interval $[-1,1]$. We emphasized this in the footnote of Lemma 5 in our revision.
>
> Q: The second one is under Lemma 5. The authors stated that "Careful singularity analysis then gives ...". It is very difficult for me to easily get these two results about Maclaurin coefficients. Could you give the detailed steps for the derivation of these results?
>
> A: Thank you for your comments. In this section, we present an overview of our proof for Theorem 1. Careful singularity analysis refers to Section 3.1 and Section 3.2. The detailed proofs for Theorem 6 and 7 in Section 3.1 and 3.2 are referred to in the Appendix. We added some clarification before Section 3.1 in our revision.
>
> Q: In Theorem 2, I am very curious about the seemingly contrary results: $H_{K_{\text{exp }}^{\gamma_1, \sigma_1}} \left(\mathbb{S}^{d-1}\right) \subseteq H_{K_{\text{exp }}^{\gamma_2, \sigma_2}}\left(\mathbb{S}^{d-1}\right)$ and $H_{K_{\mathrm{exp}}^{\gamma_2, \sigma_2}}\left(\mathbb{R}^{d}\right) \subseteq H_{K_{\mathrm{exp}}^{\gamma_1, \sigma_1}}\left(\mathbb{R}^{d}\right).$ Could you provide more explanations about this point in Section 4. The superficial explanations cannot help us understand this point. We need to go deeper into the proof sketch to figure out what makes this phenomenon happen.
>
> A: We are sorry about this confusion. Thank you for pointing it out. This is a typo in our manuscript. Both statements should be $H_{K_\text{exp}^{\gamma_2, \sigma_2}} \subseteq H_{K_\text{exp}^{\gamma_1, \sigma_1}} $, which means a smaller $\gamma$  leads to a larger RKHS. We corrected this in our revision.
>
> Thank you so much again for your detailed comments. If you have any further questions, please do not hesitate to comment.

---

### Decision · Program_Chairs · 2021-01-07
**Final Decision**

**Decision:**

Accept (Poster)

**Comment:**

The paper closes an important gap in our understanding of neural tangent kernels.
In addition, the used techniques are novel.
My low confidence is mainly based on the fact, that the review process at conference is not perfectly suited to deal with such papers, since their review would actually require both expert reviewers and substantially longer reviewing periods.